# Visual Prompt Based Personalized Federated Learning

**Guanghao Li**[*]                                                                 *lgh@nudt.edu.cn*
*National University of Defense Technology*

**Wansen Wu**[*]                                                                 *wuwansen14@nudt.edu.cn*
*National University of Defense Technology*

**Yan Sun**                                                                       *woodenchild95@outlook.com*
*The University of Sydney*

**Li Shen** [†]                                                                   *mathshenli@gmail.com*
*JD Explore Academy*

**Baoyuan Wu**                                                                    *wubaoyuan1987@gmail.com*
*The Chinese University of Hong Kong, Shenzhen*

**Dacheng Tao**                                                                   *dacheng.tao@gmail.com*
*The University of Sydney & JD Explore Academy*

**Reviewed on OpenReview:** *https://openreview.net/forum?id=dUVejidXO7*

## Abstract

As a popular paradigm of distributed learning, personalized federated learning (PFL) allows personalized models to improve generalization ability and robustness by utilizing knowledge from all distributed clients. Most existing PFL algorithms tackle personalization in a model-centric way, such as personalized layer partition, model regularization, and model interpolation, which all fail to take into account the data characteristics of distributed clients. In this paper, we propose a novel PFL framework for image classification tasks, dubbed pFedPT, that leverages personalized visual prompts to implicitly represent local data distribution information of clients and provides that information to the aggregation model to help with classification tasks. Specifically, in each round of pFedPT training, each client generates a local personalized prompt related to local data distribution. Then, the local model is trained on the input composed of raw data and a visual prompt to learn the distribution information contained in the prompt. During model testing, the aggregated model obtains client-specific knowledge of the data distributions based on the prompts, which can be seen as an adaptive fine-tuning of the aggregation model to improve model performances on different clients. Furthermore, the visual prompt can be added as an orthogonal method to implement personalization on the client for existing FL methods to boost their performance. Experiments on the CIFAR10 and CIFAR100 datasets show that pFedPT outperforms several state-of-the-art (SOTA) PFL algorithms by a large margin in various settings. The code is available at: `https://github.com/hkgdifyu/pFedPT`.

## 1 Introduction

Personalized federated learning (PFL) (Deng et al., 2020; Huang et al., 2022; Sattler et al., 2019; Shamsian et al., 2021) is a novel paradigm proposed to overcome the impacts of heterogeneity across isolated clients. Instead of training a single aggregated model like in Federated learning (FL) (Gao et al., 2022; Li et al.,

---

[*]Co-first authors
[†]Corresponding author

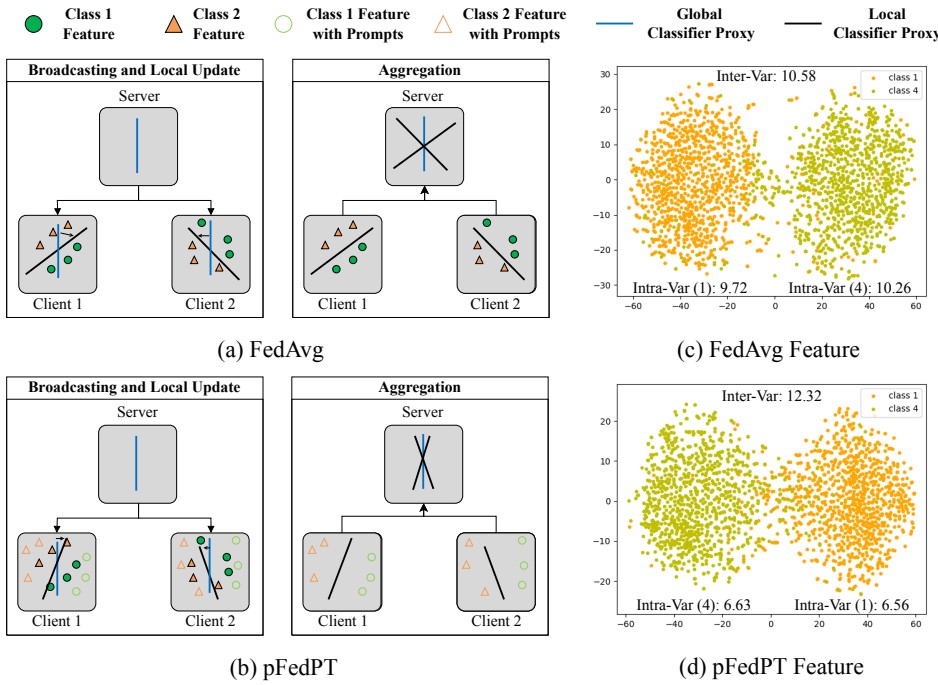

Figure 1: Differences in local update and aggregation phases between FedAvg and pFedPT. In the figure, the lines represent the decision boundaries defined by the backbone. Assume that each client has two classes represented by different shapes. (a) In FedAvg, due to the heterogeneity of data in each client, the significant difference in local updates affects the final model aggregation. The aggregation model doesn't fit well with the data on each client. However, (b) the pFedPT adds personalized visual prompts to the client data, which change the original data characteristics and improve the fit of the backbone on the client. The aggregation model can perform well on each client's data with prompt. (c) and (d) are the t-SNE visualization results of the final hidden layer trained by FedAvg and pFedPT on the client with only classes 1 and 4. pFedPT increases the inter-class variation (Inter-Var) and decreases the intra-class variation (Intra-Var).

2022; McMahan et al., 2017; Tan et al., 2022), PFL generates a personalized local model on each client that is more in line with the local data distribution by jointly considering the aggregated model and the personalized data. There are two main challenges lying in PFL. One is how to extract useful global features from models trained on each local heterogeneous dataset. The other is how to incorporate the extracted global features with the personalized features, yielding a better client-specific model.

Several works have been proposed to address the above challenges from a model perspective. PFL algorithms with a decoupling model (Arivazhagan et al., 2019; Collins et al., 2021; Oh et al., 2021) split the local model into a shared part to be aggregated with those from other clients, and a private part of maintaining locality. The shared part is used to transfer public knowledge among clients, and the private part is used to adapt to local data distribution. Clustered FL (Dinh et al., 2021) groups clients according to the similarity of the local parameters and trains an aggregated model for each group of clients. Clustered FL extracts common knowledge from similar clients within a group to generate a unified model for the group. These methods, however, still fall short in two aspects. First, these approaches rely on the effectiveness of aggregating or clustering the shared parts and may fail with highly heterogeneous data. Second, these methods simply extract the common knowledge and implement the personalization at the model level, while ignoring the potential at the data level, which may further strengthen the personalized adaptation between the aggregated model and local dataset.

In the community of computer vision (CV), both visual prompts (Liu et al., 2021) and adversarial reprogramming (Elsayed et al., 2018) employ a set of learnable parameters as a continuous task-specific vector, which can be tuned based on training data from the downstream task. Visual prompts can effectively help a large-scale pre-trained model achieve fast task transfer by simply training task-related prompts without

changing any pre-trained model parameters. The prompt parameters are like the attention guidance to implicitly hint at the task-related information for improving model performance on the new task (Liu et al., 2021). This motivates us to regard the different clients as different tasks and adopt client-specific prompts to fine-tune the aggregated model on each local client, which helps to incorporate the extracted global features with the personalized ones.

Based on this insight, we propose a novel PFL framework named pFedPT. Our approach addresses the shortcomings mentioned above by using a visual prompt to implicitly provide a hint of the data distribution on a client for the aggregated model locally. Specifically, each client model integrates a set of learnable *Prompt parameters* with a backbone participating in aggregation for classification. The prompt parameters can generate personalized visual prompts for its affiliated clients based on their local data distribution. During local training of pFedPT, the generated personalized visual prompts are added to the images. Fig. 1 (a) and (b) show the difference in the training process between FedAvg and pFedPT. For different classes of data, Fig. 1 (c) and (d) show that the generated prompts increases the inter-class variation (Inter-Var) while decreasing the intra-class variation (Intra-Var). Different class data with a visual prompt is easily distinguished by an aggregated backbone, thereby improving the local performance of the local clients. Then, the backbone is trained on the input composed of raw data and visual prompts to learn the distribution information contained in the prompt. Upon achieving convergence of the two models through alternate training, the backbone implements the extraction of common knowledge from clients and can recognize the visual prompts of different clients. The generated visual prompt reflects the client's characteristics as a client-conditional vector and implements fine-tuning of the backbone in the local client. As a result, the backbone can capture implicit knowledge about the client's data distribution based on the visual prompt and therefore obtain a better-personalized model. On the other hand, the visual prompt can be of independent interest and added as a plugin for other FL algorithms. It can fine-tune the model received by clients, which can implement the personalized improvement of the model trained by FL algorithms in different clients or further boost the performance of PFL algorithms.

We validate pFedPT on two image classification datasets, including CIFAR10 (Krizhevsky et al., 2009) and CIFAR100 (Krizhevsky et al., 2009). Empirical results show that pFedPT beats other SOTA methods of PFL with a 1%-3% improvement in test accuracy. In summary, our main contributions are four-fold:

- We propose a novel PFL framework, dubbed pFedPT, for federated image classification tasks that use the visual prompts from each client to fine-tune the aggregated model and imbue the aggregated local model with information about the local data distribution.

- pFedPT distinguishes itself with a user-oriented design that necessitates only the addition of prompt parameters. pFedPT stands out for its user-centric design. We show that pFedPT can integrate with several existing FL and decoupled PFL methods to boost their performance, which may be of independent interest.

- We conduct extensive experiments to evaluate the effectiveness of pFedPT, which significantly outperforms several SOTA baselines on CIFAR10 and CIFAR100 datasets. Besides, the experimental results illustrate that the prompt can indeed learn personalized knowledge related to the client.

## 2   Related Work

**Personalized Federated Learning (PFL).**   PFL has drawn significant research interests (Cho et al., 2021; Dai et al., 2022; Fallah et al., 2020; Hanzely et al., 2021; Li et al., 2023; Shi et al., 2023; T Dinh et al., 2020; Tan et al., 2022; Wang et al., 2023). The main difficulty of PFL is to characterize the data distributions of clients and integrate them into the federated learning training process, followed by providing a personalized local model for each client. Currently, the core idea of PFL is to decouple the model into shared layers for feature extraction and personalized layers for classification (Arivazhagan et al., 2019; Collins et al., 2021; Oh et al., 2021). Each client's parameters of the shared layer are generally updated globally using the FedAvg (McMahan et al., 2017) algorithm. In contrast, the personalized layers are trained locally and will not be shared with others. Those works focus on training a general feature extractor and a personalized classifier head for personalization.

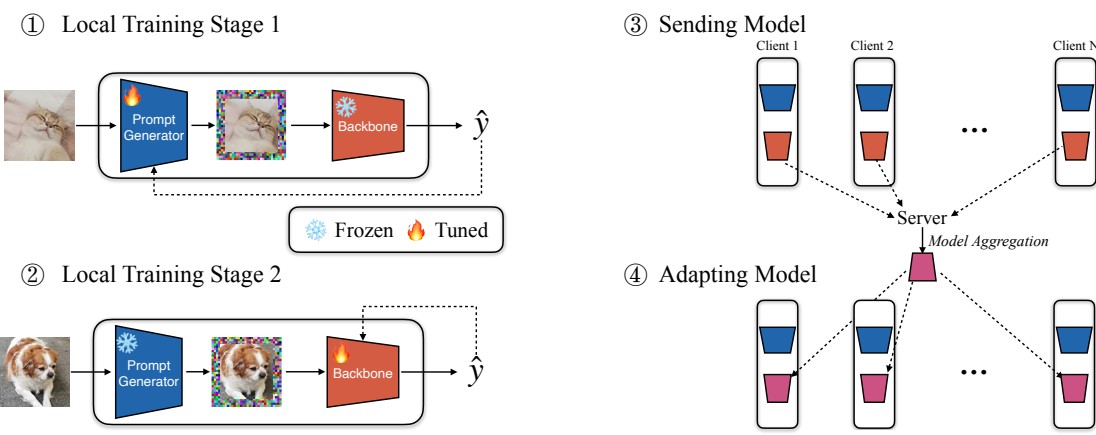

Figure 2: The pipeline of the pFedPT. $\hat{y}$ stands for the predicted logits of all classes. The dashed lines in steps 1 and 2 represent the loss backward for the model update. Each client contains a set of personalized learnable parameters preserved locally, and a Backbone, which the server will aggregate with those of other clients. The raw image input will be added to a visual prompt (colored pixels padded around the image) and then passed into the backbone for prediction.

Other work aims to combine other related machine-learning techniques with PFL. Briggs et al. (2020) and Mansour et al. (2020) use the clustering technique to divide similar clients into groups and learn a separate model for each group without inter-group federation. Dinh et al. (2021) use multitasking learning to take advantage of shared representations between clients to improve the generalization performance of each model. Yang et al. (2020) and Chen et al. (2020) use transfer learning to enhance local models by transferring knowledge between relevant clients. T Dinh et al. (2020) add regularizers to the aggregated model to prevent customers' models from simply overfitting their own data sets. Chen et al. (2018) and Fallah et al. (2020) attempt to develop a well-initialized shared aggregated model using a model-agnostic meta-learning (MAML) approach (Finn et al., 2017). In addition, fine-tuning using the aggregated model learned by the FedAvg algorithm can also improve the performance of personalized local models (Huang et al., 2023; Jiang et al., 2019). The previous works enable the model to recognize the characteristics of clients and implement personalization for them. There are also some other works (Chen et al., 2022; Goetz & Tewari, 2020; Hao et al., 2021) that focus on the data level of the client and improve the training effect of FL global model by improving the data quality on the client. However, adding additional information at the data level to achieve better performance for PFL has always been ignored. Our pFedPT framework uses visual prompts to implicitly represent the data distribution of clients, which achieves personalization by incorporating the characteristics of clients into the training process at the data level.

**Prompt Learning.** Prompt learning (Liu et al., 2021), as a novel application paradigm for large-scale pre-trained models, was first proposed in Natural Language Processing (NLP), and refers to prepending a language instruction to the original text input (Li & Liang, 2021). In this way, pre-trained models can be given hints about what tasks are currently being performed, thereby achieving strong generalization to downstream transfer learning tasks without fine-tuning the whole model (Floridi & Chiriatti, 2020). Compared to hard prompts, soft prompts avoid the trouble of manual design, and are more expressive. Lester et al. (2021) use task-specific continuous vectors as soft prompts and can be optimized by training. In the CV area, Radford et al. (2021) propose the CLIP model using language prompts to solve the vision-language tasks, which is similar to following works (Tsimpoukelli et al., 2021; Yao et al., 2021). In (Bahng et al., 2022), the visual prompts are designed as an input-agnostic perturbation, which is padded around the input images. The perturbation-generating function includes a small number of trainable parameters, which helps the pre-trained vision models perform downstream tasks without fine-tuning any parameters. Visual Prompt Tuning (VPT) (Jia et al., 2022) is introduced as a parameter-efficient alternative to full fine-tuning for pre-trained model (Dosovitskiy et al., 2021).

Two contemporary studies, PROMPTFL (Guo et al., 2022) and FedPrompt (Zhao et al., 2022), also integrate prompt learning into Federated Learning (FL). However, our approach, pFedPT, exhibits several key differences from these works. A succinct comparison elucidates these distinctions: **(i) FL Category:** pFedPT is specifically tailored for personalized FL, focusing on individual client needs. Conversely, PROMPTFL and FedPrompt are designed for generic FL, aiming for broader applicability without client-specific customization. **(ii) Prompt Deployment:** In pFedPT, prompts are retained locally at each client, cultivating a unique, personalized prompt for every participant. This contrasts with PROMPTFL and FedPrompt, where prompt parameters are shared among all clients, culminating in a universally applicable FL model. **(iii) Training Dynamics:** pFedPT diverges significantly in its training methodology. Unlike PROMPTFL and FedPrompt, which freeze the pre-trained model and only update prompt parameters, pFedPT dynamically trains both backbone and prompt parameters. This alternating training regimen enhances model personalization and adaptability to specific client data. These distinctions underscore pFedPT's commitment to personalization in FL, setting it apart from PROMPTFL and FedPrompt's more generalized approach.

## 3 Methodology

In this section, we introduce the proposed visual prompt based personalized federated learning (pFedPT) framework. Below, we first provide several preliminaries on PFL.

### 3.1 Problem setup

Suppose that there are $N$ clients, denoted as $C_1, ..., C_N$, respectively. Client $C_i$ has a local dataset $\mathcal{D}^i$ with $|\mathcal{D}^i|$ samples. The goal of traditional FL (McMahan et al., 2017) is to collaboratively learn a machine learning model $w$ over the dataset $\mathcal{D} \triangleq \bigcup_{i \in [N]} \mathcal{D}^i$ with the help of a central server, while the raw data are not exchanged. The objective of FL is defined below:

$$\arg\min_w \mathcal{L}(w) = \sum_{i=1}^{N} \frac{|\mathcal{D}^i|}{|\mathcal{D}|} L_i(w), \tag{1}$$

where $L_i(w) = \mathbb{E}_{(x,y) \sim \mathcal{D}^i}[\ell_i(w; (x, y))]$ is the empirical loss of $C_i$. However, rather than aiming at a single aggregated model $w$ in FL, PFL is supposed to train personalized models $w_i$ for different clients (Tan et al., 2022), which is defined as the following optimization problem:

$$\arg\min_W \mathcal{L}(W) = \sum_{i=1}^{N} \frac{|\mathcal{D}^i|}{|\mathcal{D}|} L_i(w_i), \tag{2}$$

where $W = \{w_1, ..., w_N\}$ is the personalized models set for all clients.

### 3.2 Workflow of pFedPT

We introduce a novel visual prompt based PFL framework for solving the PFL task, dubbed pFedPT. The central insight of the pFedPT is to train learnable continuous visual prompts about data distribution for each client and use them to fine-tune backbones locally on those clients. Prompts on each client is client-specific knowledge aiding the backbone to complete the training task. The pFedPT currently focuses on visual-related tasks, wherein each client maintains a set of prompt parameters and a backbone, as shown in Fig. 2. When performing image classification tasks, pFedPT first adds prompts to each image, which is then passed into the backbone for classification prediction. Generally, a complete pFedPT training process mainly includes four steps, as shown in Fig. 2:

- **Step 1.** To begin with, the parameters of the prompt on each client are updated with local data while the whole backbone is frozen.

- **Step 2.** After training several epochs, the prompt parameters will be frozen, and the backbone will begin to update for a fixed number of epochs.

- **Step 3.** When the training process of all clients is finished, they send the trained backbone to the server, followed by the aggregation operation conducted by the server.

- **Step 4.** The aggregated backbone will be broadcast to every client to replace the old backbone stored locally.

Repeat the **Step 1-Step 4** until the training process of the prompt parameters and backbone converges. At this point, the prompt parameters for each client are based on local data distribution and can be seen as a guide to fine-tuning the prediction results of the backbone for the input images. Since the prompt is client-specific, the same backbone can generate different fine-tuning effects when used by different clients to achieve personalization.

Below, we specify the key components of pFedPT, i.e., prompt parameters, which are parameterized with the parameter $\delta$ for the prompt. The prompt is added to the input image to form a prompted image $X_i + \delta_i$. During the local evaluation, the optimized prompt is added to all test images,

$$\mathcal{X}_i = \left\{ x_i^1 + \delta_i, \ldots, x_i^n + \delta_i \right\}. \tag{3}$$

There are several ways to design a visual prompt in terms of template and size. Following the settings of (Bahng et al., 2022), we explore three visual templates: pixel patch at a random location, pixel patch at a fixed location, and padding. We explore various prompt sizes $p$, where the actual number of parameters is $Cp^2$ for patches and $2Cp(H + W - 2p)$ for padding, where $C, H, W$ are the image channels, height, and width, respectively. In order to explore the effect of different prompts on the results, we conducted an experiment on CIFAR10 dataset with a Dirichlet (0.3) partition. Fig. 8 shows that padding prompts with $p = 4$ size achieve the best performance over other design choices. We use this as the default for all our experiments.

### 3.3 Modeling for pFedPT

Our goal is to learn a personalized prompt $\delta_i$ for each client and a backbone $w$. The prompt $\delta_i$ is also trained by the local data. Our objective is to solve the following:

$$\underset{w, \delta_i}{\arg \min} \mathcal{L}(w, \delta_i) = \sum_{i=1}^{N} \frac{|\mathcal{D}^i|}{|\mathcal{D}|} L_i(w, \delta_i), \tag{4}$$

where $L_i(w, \delta_i) = \mathbb{E}_{(x,y) \sim \mathcal{D}^i}[\ell_i(w; (x + \delta_i, y))]$ is the empirical loss of $C_i$. To achieve the goal in Eq. (2), existing PFL algorithms usually add a regularizer to the model to perform information exchange between clients (Arivazhagan et al., 2019; Li et al., 2021b), partition the layers as shared and personalized parts by exchanging the shared layers (Tan et al., 2022), or interpolate the aggregated model with local models (Li et al., 2021b; Mansour et al., 2020). However, pFedPT still uses the aggregated model $w$ to deliver public knowledge between clients, and personalized knowledge is incorporated by adding $\delta_i$ to the data. Specifically, the shared backbone is responsible for the extraction of the common knowledge of each client and identifying the information carried by the visual prompt of the individual clients. The client-specific prompt is responsible for increasing the guidance of the backbone to achieve finze-tuning to adapt to the client's data distribution. We implement personalized prediction of the backbone at the client data level.

### 3.4 Optimization for pFedPT

To achieve the optimization goal of Eq. (4), we alternately update the prompt parameters and the backbone on each client using gradient descent. pFedPT first trains the prompt parameters with the aggregation model fixed, and the model maximizes the likelihood of the correct label $y$, which is equivalent to solving:

$$\underset{\delta_i}{\arg \min} \mathcal{L}_i(w_i, \delta_i) = \mathbb{E}_{(x,y) \sim \mathcal{D}^i}[\ell_i(w_i; (x + \delta_i, y))]. \tag{5}$$

After updating the prompt parameters locally, we freeze the parameters of the prompt parameters, and then train the backbone for several epochs. The backbone has the following objective function in the client $i$

---

**Algorithm 1:** pFedPT framework

---

**Input:** number of communication rounds $T$, the set of clients $\{C_1, ..., C_N\}$, number of local epochs $E_b$ for backbone, number of local epochs $E_g$ for the prompt parameters, learning rate $\eta_b$ for backbone, learning rate $\eta_g$ for the prompt parameters, initialization parameters $w^0$ for backbone, initialization parameters $\delta_i^0$ for the prompt parameters in client $i$.

**Output:** The final model $w^T$

**1 Server executes**: initialize $w^0$

**2** $\mathcal{S}$ : choose a random set of devices from $C$

**3 for** $t = 0, 1, ..., T - 1$ **do**

**4**      **for** $C_i \in \mathcal{S}$ **in parallel do**

**5**          send the aggregated model $w^t$ to $C_i$

**6**          $w_i^t \leftarrow$ **LocalTraining**$(i, w^t)$

**7**      $w^{t+1} \leftarrow \sum_{i=1}^{\|\mathcal{S}\|} \frac{|\mathcal{D}^i|}{|\mathcal{D}|} w_i^t$

**8 return** $w^T$

**9 LocalTraining**$(i, w^t)$: $w_i^t \leftarrow w^t$

**10 for** epoch $i = 1, 2, ..., E_g$ **do**

**11**      **for** each batch $\mathbf{b} = \{x, y\}$ of $\mathcal{D}^i$ **do**

**12**          *Training for prompt parameters:* $\delta_i^t \leftarrow \delta_i^t - \eta_g \nabla \ell_i(w_i^t; (x + \delta_i, y)))$

**13 for** epoch $i = 1, 2, ..., E_c$ **do**

**14**      **for** each batch $\mathbf{b} = \{x, y\}$ of $\mathcal{D}^i$ **do**

**15**          *local backbone training:* $w_i^t \leftarrow w_i^t - \eta_b \nabla \ell_i(w_i^t; (x + \delta_i, y))$

**16 return** $w_i^t$ to server

---

during the training process:

$$\arg \min_w \mathcal{L}_i(w_i, \delta_i) = \mathbb{E}_{(x,y) \sim \mathcal{D}^i}[\ell_i(w_i; (x + \delta_i, y))]. \tag{6}$$

A locally trained backbone can learn the client data distribution corresponding to a prompt on the client and prompt knowledge is passed between clients via model aggregation at the server. The backbone on the server aggregates according to the following formula:

$$w^{t+1} \leftarrow \sum_{i=1}^{N} \frac{|\mathcal{D}^i|}{|\mathcal{D}|} w_i^t, \tag{7}$$

where $t$ represents the number of training rounds. We summarize the detailed procedures of pFedPT in Algorithm 1.

In the end, we give several comments on the differences between our pFedPT and decoupled FedRep (Collins et al., 2021). Fig.3 describes their training process. Note that pFedPT also has a private part and a public part, but the private part is the prompt parameters that we added at the client data level additionally. The personalized visual prompt generated adds the client's personalized knowledge to the training process by fine-tuning backbone's input without changing backbone's inference process. FedRep is to separate the private part in the inference model, and different clients have different inference processes. The objective functions of FedRep and pFedPT are also different. Furthermore, the visual prompt is orthogonal to FedRep type methods, which can be integrated together to further boost their performance.

## 4 Experiments

In this section, we evaluate the effectiveness of pFedPT and compare it with several advanced methods in various datasets and settings. We also conduct a number of exploratory experiments to find out how pFedPT works and verify the effectiveness of pFedPT in terms of client data distribution. The aim of our experiments is to address the following research questions:

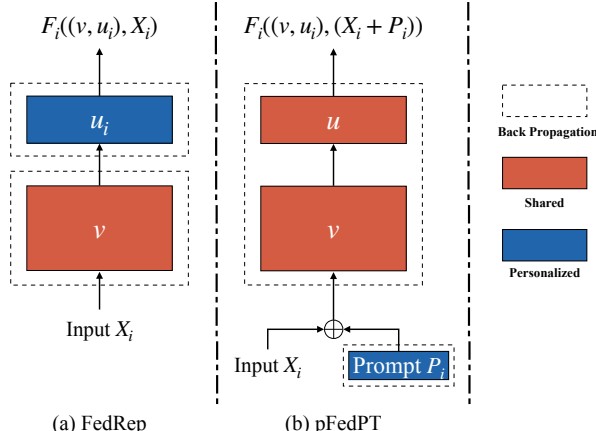

Figure 3: Differences between pFedPT and decoupled personalized FL algorithm (FedRep)

- **RQ1** Can the proposed pFedPT achieve better performance than other methods?

- **RQ2** Is pFedPT robust and performs well in different situations?

- **RQ3** Whether pFedPT can be combined with other algorithms to improve performance?

- **RQ4** Whether the prompt that is added to the image is noticed by the backbone model?

- **RQ5** Is the information contained in the prompt different from client to client?

- **RQ6** Does prompt contain information about client data distribution?

- **RQ7** Whether the parameters of prompts can converge?

- **RQ8** What are the effects of different types of prompts?

## 4.1 Experimental Setup

**Comparison methods.** We compare pFedPT with several advanced FL methods. FedAvg (McMahan et al., 2017) is proposed as the basic framework in federated learning. FedProx (Li et al., 2020) adds a proximal term to the objective function of the local model and allows for the emergence of incomplete training of the local model. MOON (Li et al., 2021a) is to utilize the similarity between model representations to correct the local training of individual parties, conducting contrastive learning at the model level. Fine-tuning of the global model has been demonstrated to be highly effective in PFL (Chen & Chao, 2021; Yu et al., 2020), therefore, we apply fine-tuning to the global models learned using FedAvg, FedProx, and MOON. Per-FedAvg (Fallah et al., 2020) proposes a personalized variant of FL in a heterogeneous setting and introduces the PerFedAvg algorithm as a decentralized implementation to solve the problem. FedPer (Arivazhagan et al., 2019) and FedRep (Collins et al., 2021) are base + personalization layer approaches for federated training of deep feed-forward neural networks, which can combat the ill-effects of statistical heterogeneity. FedMTL (Smith et al., 2017) uses a multi-task learning (MTL) framework to learn separate models for each client. FedBABU (Oh et al., 2021) achieves good personalization performance by freezing the last discriminative layer of the network and fine-tuning it after training. We also compare a baseline named Local, where each client trains a model with its local data without federated learning.

**Datasets.** We conduct experiments on two benchmark datasets: CIFAR10 (Krizhevsky et al., 2009) and CIFAR100 (Krizhevsky et al., 2009). The CIFAR10 dataset contains 50,000 training data and 10,000 test data in 10 classes. Each data sample is a $3 \times 32 \times 32$ color image. CIFAR100 (Krizhevsky et al., 2009) includes 50,000 training data and 10,000 test data in 100 classes as 500 training samples per class. CIFAR100 is a more difficult dataset for classification tasks than CIFAR10. For CIFAR10 and CIFAR100, we normalize the pixel value within a specific mean and std value in our code, which are [0.5, 0.5, 0.5] for the mean and

[0.5, 0.5, 0.5] for the std.We consider two different scenarios for simulating non-identical data distributions (Non-IID) across federated clients. Dirichlet Partition follows works (Hsu et al., 2019), where we partition the training data according to a Dirichlet distribution $Dir(\alpha)$ for each client and generate the corresponding test data for each client following the same distribution. We specify $\alpha$ equal 0.3 for each dataset. In addition, we evaluate with the pathological partition setup similar to (Zhang et al., 2020), in which each client is only assigned a limited number of classes at random from the total number of classes. We specify that each client possesses 5 classes for CIFAR10 and 50 classes for CIFAR100.

**Evaluation Metrics.** We are distributing the test data set to each client in the same way as the training set and the final accuracy of each method reported in our results is the average accuracy of the local model for each client on its own test set after training.

**Implementation Details.** We verify the experimental results based on CNN and ViT architectures. The CNN model consists of 2 convolutional layers with 64 5×5 filters followed by 2 fully connected layers with 394 and 192 neurons and a softmax layer. We use tiny ViT architecture consisting of 8 blocks with 8 self-attention layers in each block. The corresponding attention head number is 8, the patch size is 4, and the embedding dimension is 128. We set the number of clients to 50, and then each client has a 20% chance of participating in each communication round. We utilize the SGD algorithm (Cherry et al., 1998) as the local optimizer for all methods. We use padding as our prompt method. We set batch size as 16 in the local training phase, the local training epochs for the prompt parameters and backbone as 5 in each round, the learning rate for the backbone as 0.005, the learning rate for the prompt parameters as 1, and the padding prompt size as 4. The number of communication rounds is set to 150 for CIFAR10, 300 for CIFAR100, where all FL approaches have very limited or no accuracy gain with more communications.

**Hyper-parameters Settings.** We fix the learning rate for local training as 0.005 and for the prompt parameters training as 1.0. We fix the training batch size as 16 and fix the epoch for local training as 5. For the specific parameters in FedProx, the proximal rate is set as 0.0001. For the specific parameters in MOON, the $\mu$ is set as 1.0. For the specific parameters in FedRep, the personalized learning rate is set as 0.01. For the specific parameters in FedMTL, the iterations for solving quadratic sub-problems are set as 4000. For the specific parameters in FedBABU, the fine tuning step is set as 1. For fine-tuning, we train the global model learned in FedAvg, FedProx, and MOON for 5 epochs.

Table 1: The results of pFedPT and baseline methods on the image datasets with different non-IID settings. * denotes the fine-tuning results.

| #setting | CIFAR10 | | | | | | CIFAR100 | | | | | |
| | IID | | Dirichlet | | Pathological | | IID | | Dirichlet | | Pathological | |
| #client | ViT | CNN | ViT | CNN | ViT | CNN | ViT | CNN | ViT | CNN | ViT | CNN |
|---|---|---|---|---|---|---|---|---|---|---|---|---|
| FedAvg | 60.50 | **67.13** | 53.01 | 61.92 | 54.98 | 63.62 | 29.60 | 26.42 | 25.93 | 26.50 | 27.71 | 30.28 |
| FedAvg* | 60.47 | 67.12 | 72.43 | 77.52 | 67.27 | 74.95 | 29.97 | 25.75 | 35.88 | 32.51 | 34.79 | 31.84 |
| FedProx | 57.04 | 66.94 | 53.14 | 61.95 | 55.02 | 63.29 | 27.71 | 26.29 | 26.00 | 26.48 | 27.84 | 30.52 |
| FedProx* | 57.05 | 66.82 | 73.94 | 77.43 | 68.54 | 74.90 | 27.43 | 25.51 | 36.54 | 32.17 | 34.22 | 35.77 |
| MOON | 60.99 | 66.88 | 61.12 | 62.53 | 65.98 | 63.52 | 29.32 | **26.43** | 24.95 | 26.93 | 27.61 | 29.00 |
| MOON* | 60.06 | 66.31 | 73.46 | 78.69 | 70.12 | 75.10 | 29.41 | 25.31 | 36.20 | 30.96 | 33.74 | 36.02 |
| Per-FedAvg | 57.48 | 61.35 | 68.63 | 76.81 | 74.56 | 72.32 | 26.41 | 26.12 | 31.67 | 30.17 | 34.65 | 32.38 |
| FedPer | **61.57** | 51.46 | 73.16 | 77.98 | 75.20 | 79.97 | 29.74 | 10.82 | 36.78 | 27.79 | 35.36 | 31.13 |
| FedRep | 48.38 | 49.70 | 74.11 | 77.65 | 74.48 | 78.39 | 17.84 | 9.13 | 35.06 | 27.39 | 36.13 | 32.41 |
| FedMTL | 45.65 | 45.65 | 68.48 | 73.95 | 65.39 | 70.94 | 17.91 | 7.34 | 26.08 | 25.85 | 25.46 | 26.32 |
| FedBABU | 50.41 | 61.17 | 74.21 | 80.11 | 74.30 | 80.69 | 20.61 | 22.55 | 36.17 | 31.66 | 35.74 | 35.45 |
| Local | 45.37 | 39.04 | 68.40 | 73.98 | 64.83 | 70.76 | 18.01 | 7.33 | 26.23 | 25.15 | 24.65 | 25.34 |
| pFedPT (ours) | 60.01 | 66.09 | **74.92** | **80.83** | **75.42** | **81.16** | **31.66** | 26.41 | **36.80** | **32.47** | **36.88** | **37.98** |

## 4.2 Main Results

We run vast experiments to determine the superiority of pFedPT on the model performance in different datasets. Our results highlight the benefit of pFedPT compared to the existing PFL optimization approaches.

**Comparisons with SOTA methods.** Tab. 1 presents a comparative analysis of pFedPT's best accuracy against baseline methods on evaluation datasets under various settings, addressing **RQ1**. On CIFAR10 and CIFAR100 datasets, especially in Non-IID settings, pFedPT consistently registers the highest test accuracy. For example, with a CNN trained on CIFAR10 data following a Dirichlet distribution, pFedPT achieves a test accuracy of 80.83%, surpassing FedAvg's 61.92%, FedPer's 77.98%, and even FedAvg with fine-tuning at 77.52%. This demonstrates that, while fine-tuning can enhance model accuracy, pFedPT excels beyond these methods. The superior performance of pFedPT can be attributed to the effective utilization of prompts in each client, significantly boosting the backbone model's performance. In the CIFAR100 dataset as well, pFedPT outshines most baseline methods across different settings and delivers competitive results in the Dirichlet setting. More experimental results on Tiny-ImageNet can be found in the Appendix A.2.

Table 2: Architecture Agnostic analysis. We insert the proposed framework (denote as PT) to existing pFL methods.

| #setting | CIFAR10 | | | CIFAR100 | | |
|---|---|---|---|---|---|---|
| | IID | Dirichlet | Pathological | IID | Dirichlet | Pathological |
| FedProx | 66.94 | 61.95 | 63.29 | 26.29 | 26.48 | 30.52 |
| FedProx+PT | **67.60** | **80.47** | **81.48** | **26.87** | **31.95** | **37.88** |
| MOON | 66.88 | 62.53 | 63.52 | 26.43 | 26.93 | 29.00 |
| MOON+PT | **66.92** | **77.84** | **76.00** | **26.58** | **28.67** | **34.60** |
| FedPer | 51.76 | 77.98 | 79.97 | 10.82 | 27.79 | 31.13 |
| FedPer+PT | **52.02** | **78.40** | **80.59** | **11.37** | **28.83** | **31.14** |
| FedRep | 49.70 | 77.65 | 78.39 | 9.13 | 27.39 | 32.41 |
| FedRep+PT | **49.88** | 77.65 | **79.11** | **9.87** | **29.19** | **32.75** |

**Robustness of pFedPT.** Our pFedPT demonstrates significant success in both ViT and CNN models. It exhibits enhanced performance as the FL tasks become increasingly challenging with a greater Non-IID data distribution, effectively addressing **RQ2**. In the IID setting, however, we observe a performance degradation in all personalized solutions when compared to the FedAvg approach. Our interpretation of this phenomenon is that, in the IID setting, data across different clients closely mirror the overall data distribution. Consequently, the effectiveness of many PFL methods is diminished, with FedAvg frequently outperforming most PFL approaches in such contexts. pFedPT leverages client-specific data distribution information through visual prompts. In scenarios where data is IID, the output across various clients tends to converge, resulting in performance akin to that of FedAvg.

**Improvements of prompt for other algorithms.** To answer **RQ3**, we combine PfedPT with other methods and find visual prompts can improve the performance of backbones on clients by fine-tuning the backbone with prompts about the distribution of the client's data. We explore the usefulness of visual prompts as client-specific knowledge for other FL algorithms, and Tab. 2 presents these results. In the Dirichlet setting of CIFAR10, the final test accuracy of FedProx increases from 61.95% to 80.47% after adding prompts, and the test accuracy of MOON increases from 62.53% to 77.84%. We find that a visual prompt enables fine-tuning of the backbone of the client, which helps FL algorithms that pursue high precision fuse client information for personalization. Similarly, PFL algorithms with model decoupling, like FedRep and FedPer, can also yield a performance boost by integrating pFedPT. Therefore, prompt can be used as an additional component to improve the personalization performance of some existing FL algorithms.

Compared with other baselines, pFedPT takes full advantage of the data improvement space. Additional prompts are added to the data entered into the model to improve the performance of each client.

### 4.3 Exploratory Study

To provide more explanation for pFedPT, we additionally conduct several exploratory studies on pFedPT.

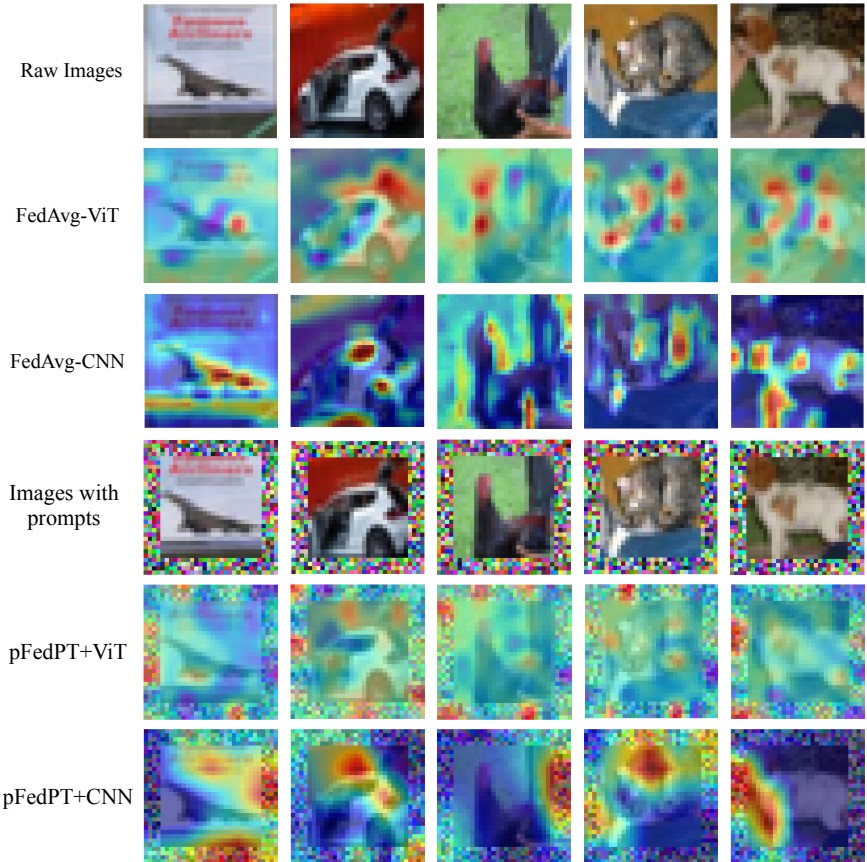

Figure 4: Visualization results generated by FedAvg and pFedPT with different backbones.

**Visualization of attention maps.** To illustrate the effectiveness of visual prompts and answer **RQ4**, we conducted some validation experiments. We train ten clients using FedAvg and pFedPT with ViT and CNN backbones under the Dirichlet setting of the CIFAR10 dataset, respectively. As shown in Fig. 4, we make a visualization of the attention map of the last layer in the ViT and CNN by Grad-CAM (Selvaraju et al., 2020). The first three rows in the figure show that FedAvg focuses on some salient classification features of the raw image. The fourth row contains the input images with the padding visual prompts, which are added by the prompt parameters of pFedPT according to Eq. (3). Both pFedPT+ViT and pFedPT+CNN shift some attention to the added prompts, which can help obtain the client-specific knowledge for the model inference process, thus improving the performance of the model.

**The guidance information contained in the prompts.** In order to further explore the influence of visual prompts and answer **RQ5**, we generated 100 different pure color images with the shape of $[3 \times 32 \times 32]$. Using the pure color picture, pFedPT can exclude the disturbance of image contents and pay more attention to visual prompts. We feed those color pictures into pFedPT models in different clients with different prompts and visualize the output embeddings of their last MLP layer. We project them into a two-dimensional plane using the t-SNE algorithm (Van der Maaten & Hinton, 2008). Fig. 5 illustrates that with the incorporation of visual prompts, the representation space of a given client becomes distinct from that of other clients. Although this difference may require the backbone to exert more effort to align on the server, it enables the model outputs of various clients to be easily distinguished. This indicates that the prompts encapsulate client-specific knowledge within the client model, thereby significantly facilitating the classification task.

**The connection between Prompt and the client data distribution.** To answer answer **RQ6** we use a single, pure-color image as input to investigate the relationship between the local model output and the data distribution of each client. Ideally, the output distribution over classes of each pFedPT client should

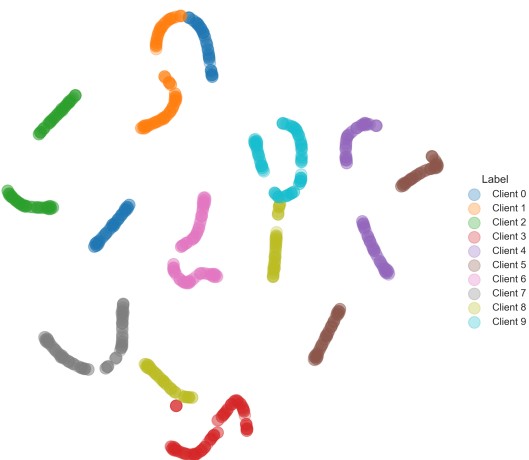

Figure 5: t-SNE visualization of embedding for pure color images with learned prompts in different clients.

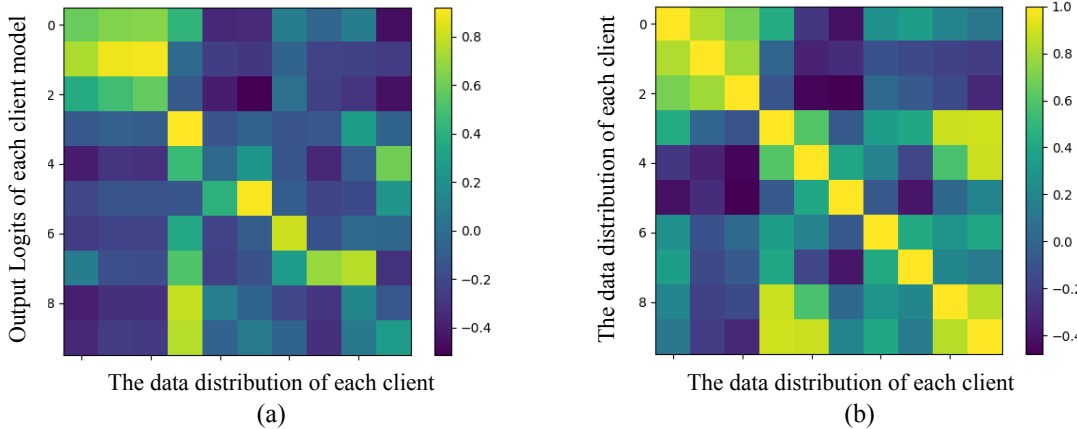

Figure 6: Similarity comparison between the distribution of the predicted classes and the local data.

align with the local data distribution. Fig. 6 reveals that after adding the visual prompts, the outputs of the pFedPT will be similar to the distribution of the client itself. The difference between the visual prompts generated by clients with similar data distribution is also smaller, which means that the visual prompts indeed contain the data distribution information of the clients. Therefore, the visual prompts provide the model with certain client-specific knowledge when classifying a specific client and assist in the classification task.

**Empirical analysis of the learned prompts.** Fig. 7 records the average difference of the prompts generated between the two rounds before and after ten clients during the pFedPT training process. The overall experimental results are divided into two stages: first ascending and then descending. In our settings, the initial prompt parameters of each client are the same, and the rising stage is the mapping process between each client and the prompts based on its own data distribution. The descending stage is when the aggregated model tends to converge, and the mapping between the prompt and the client data distribution on each client is complete. Eventually, the change in prompt embedding approaches 0, and each client establishes stable prompts that conform to its own data distribution which answer **RQ7**.

**Impact of different types of visual prompts.** We analyze different choices on how and where to insert prompts in the input images and how they would affect the final performance to answer **RQ8**. We perform an ablation study on different prompt sizes in $p = \{2, 4, 6, \ldots, 16\}$ in CIFAR10 with a Dirichlet distribution. As shown in Fig. 8, padding prompts reach the highest performance with a size of 4. The test accuracy of fixed location and random location prompts grows gradually with the increase in prompt size, but it is

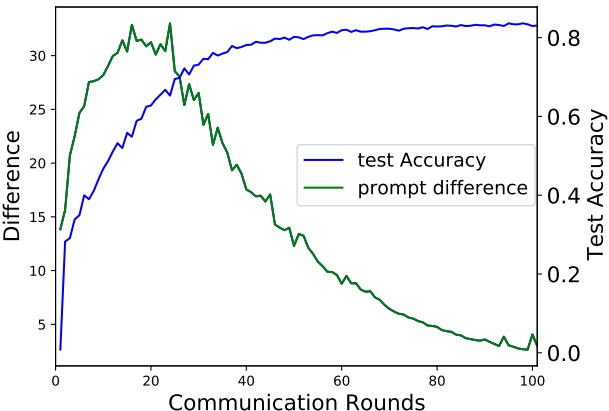

Figure 7: The difference of prompt between two consecutive rounds.

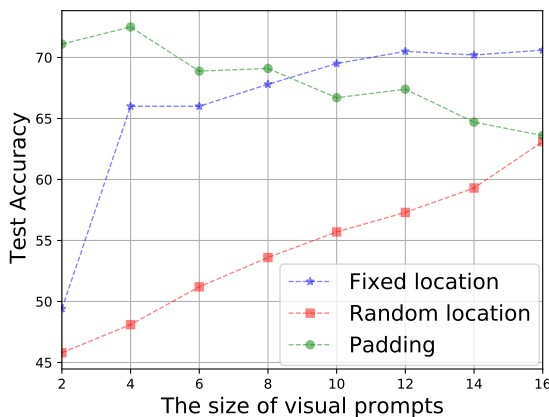

Figure 8: Effect of different types of prompts

still slightly lower than the padding prompt. In contrast, the accuracy of padding prompts decreases as the prompt size increases. A possible explanation is that the padding method covers more pixels of the original images than the other two methods when using the same length of prompts. As a result, the key information for classification could be blocked by the prompts and harm the performance of the model. Overall, the padding prompts with size 4 achieve the best performance. Note that other visual tasks may require significantly different kinds of prompts.

## 5 Conclusion

In this work, we propose a novel framework named pFedPT, a personalized federated learning method based on visual prompts. We make the first attempt to introduce visual prompts to personalized federated learning, using a set of prompt parameters to distill information from local data into the visual prompts and fine-tune the backbone. In the process of pFedPT training, the backbone could use the guidance information from visual prompts to perform the personalized downstream tasks. Since the prompt parameters is trained locally on the client, it does not reveal data distribution information about the client to others or the server. pFedPF can also serve a strong plugin to boost the performance of existing FL methods, which could be of independent interest. We provide extensive experiments to illustrate how the pFedPT works and demonstrate its effectiveness in experiments with heterogeneous settings and several types of dataset partition.

## Acknowledgment

This work is supported by STI 2030—Major Projects (No. 2021ZD0201405). I would like to express my sincere gratitude to the contributors of the GitHub repository `PFLlib` (`https://github.com/TsingZ0/PFLlib`). Their work laid the foundation for my research and significantly contributed to the success of this work.

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

## A Appendix: More Experiment Results

We run experiments on the real-world datasets for image classification tasks, including CIFAR10, CIFAR100, and Tiny ImageNet. We conduct comprehensive investigations for the impact on client heterogeneity by designing IID and non-IID data scenarios. For comparison, we utilize the FedAvg, FedProx, and FedPer algorithms as baselines. The experiment settings are described in detail below.

### A.1 Setups

**Dataset.** We adopt real-world datasets for the image classification task, including CIFAR10, CIFAR100, and Tiny ImageNet. The CIFAR10 dataset contains 50,000 training data and 10,000 test data in 10 classes. Each data sample is a $3 \times 32 \times 32$ color image. CIFAR100 (Krizhevsky et al., 2009) includes 50,000 training data and 10,000 test data in 100 classes as 500 training samples per class. TinyImageNet (Oord et al., 2018) involves 100,000 training images and 10,000 test images in 200 classes for $3 \times 64 \times 64$ color images, as shown in Table 3. For CIFAR10/100 and TinyImageNet, we normalize the pixel value within a specific mean and std value in our code, which are [0.5, 0.5, 0.5] for the mean and [0.5, 0.5, 0.5] for the std.

Table 3: The similarity between predicted and real data distribution

| Datasets | Training Data | Test Data | Class | Size |
|---|---|---|---|---|
| CIFAR-10 | 50,000 | 10,000 | 10 | $3 \times 32 \times 32$ |
| CIFAR-100 | 50,000 | 10,000 | 100 | $3 \times 32 \times 32$ |
| TinyImageNet | 100,000 | 10,000 | 200 | $3 \times 64 \times 64$ |

**Backbone.** We adopt two backbones including ViT and CNN for experiments. Given an image $I \in \mathbb{R}^{3 \times h \times w}$, the ViT reshapes it to a sequence of flattened 2D patches $I_p \in \mathbb{R}^{n \times (p^2 \cdot c)}$, where $c$ is the number of channels. $(h, w)$ is the height and width of the original image, while $(p, p)$ is the size of each image patch. A trainable linear projection flattens the patches into a latent $D$-dimensional embedding space, which is then embedded with a positional embedding. The transformer encoder consists of $N$ layers of Multi-Head Self-Attention (MSA) and Multi-Layer Perceptron (MLP) blocks. For MSA, the queries, keys and values are generated via linear transformations on the inputs for $K$ times with one individual learned weight for each head. Then in parallel, the attention function is applied to all queries, keys, and values. The sequence of image patches $I_p$ is passed into the MSA, followed by MLP for $N$ times to get the final outputs.

The ViT used by us consists of 8 blocks with 8 self-attention layers in each block. The corresponding attention head number is 8, the patch size is 4 and the embedding dimension is 128. The CNN consists of the basic modules of CNN, including two conventional layers with 64 of $5 \times 5$ convolution kernels, each conventional layer followed by a down-pooling larger, after that are two fully connected layers with 394 and 192 neurons and a softmax layer for prediction.

**Three types of generating Visual Prompts.** Following the settings of (Bahng et al., 2022), we explore three visual templates: pixel patch at a random location, pixel patch at a fixed location, and padding. We describe in Fig. 9 the difference between different visual prompts and how these visual prompts templates are added to the input picture. The comparison results are shown in Fig. 8.

**Dataset Partitions.** To fairly compare with the other baselines, we introduce heterogeneity by splitting the total dataset and sampling the label ratios from the Dirichlet distribution and Pathological distribution. An additional parameter is used to control the level of heterogeneity of the entire data partition. In order to visualize the distribution of heterogeneous data, we make heat maps of the label distribution in different datasets, as shown in Fig. 10. It could be seen that for heterogeneity weight equal to 0.3 in the Dirichlet distribution, about 10% to 20% of the categories dominate on each client, which is the blue block in Fig. 10. For heterogeneity weight equal to 5 in Pathological distribution, 50% of the categories dominate on each client, which is the black block in Fig. 10. The IID dataset is totally averaged for each client, which is the green block in Fig. 10.

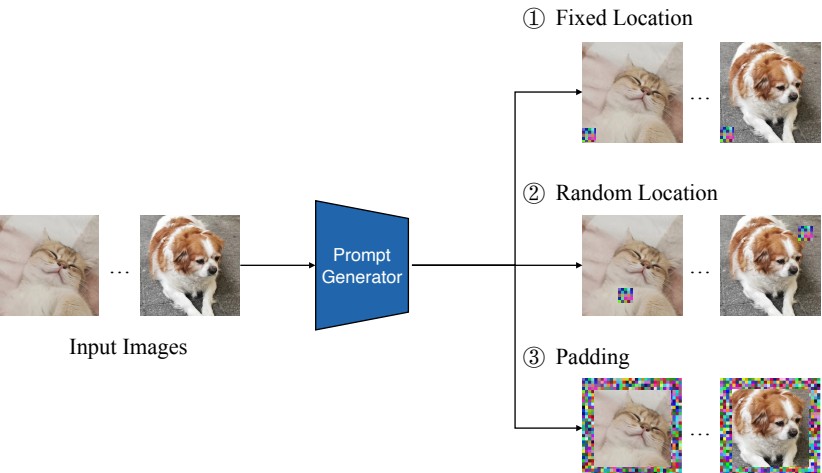

Figure 9: Different visual prompt templates

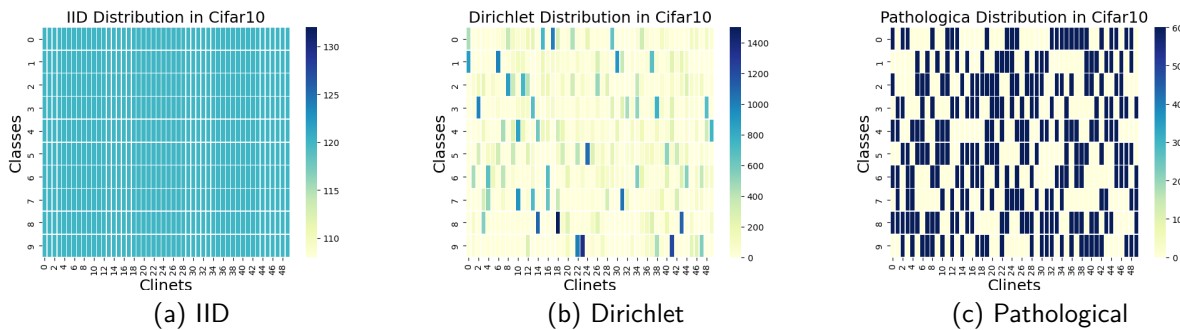

| (a) IID | (b) Dirichlet | (c) Pathological |

Figure 10: Heat maps for each client with CIFAR10 dataset under different data partitions. The color bar denotes the number of data samples. Each rectangle represents the number of data samples of a specific class in a party.

## A.2 Other Experimental Results

Table 4: The results of the pFedPT and the benchmark methods on the TinyImageNet dataset with different non-IID settings and backbones.* denotes the fine-tuning results.

| #setting | TinyImageNet | | | | | |
| | IID | | Dirichlet | | Pathological | |
| #client | VIT | CNN | VIT | CNN | VIT | CNN |
|---|---|---|---|---|---|---|
| FedAvg | 15.47 | 16.22 | 12.06 | 11.01 | 9.37 | 10.53 |
| FedAvg* | 16.82 | 16.98 | 21.32 | 17.83 | 19.25 | 18.02 |
| FedProx | 15.53 | 15.81 | 13.82 | 10.98 | 12.84 | 11.32 |
| FedProx* | 15.62 | 16.21 | 21.89 | 18.85 | 20.48 | 18.32 |
| MOON | 15.20 | 16.78 | 15.51 | 11.05 | 13.67 | 10.24 |
| MOON* | 15.84 | **17.08** | 22.33 | 18.96 | 19.68 | 17.85 |
| FedPer | 14.85 | 13.18 | 23.95 | 20.21 | 19.23 | 17.99 |
| FedRep | 12.91 | 11.56 | **25.24** | 21.32 | 23.43 | 20.42 |
| FedMTL | 9.75 | 11.02 | 21.14 | 17.96 | 18.30 | 17.39 |
| FedBABU | 13.61 | 12.04 | 24.34 | 18.62 | 22.46 | 18.41 |
| Local | 9.82 | 10.72 | 20.86 | 17.43 | 18.19 | 17.12 |
| pFedPT (ours) | **18.72** | 16.53 | 21.21 | **21.42** | **25.95** | **20.66** |

**Experimental results for TinyImageNet.** We compare the performance of pFedPT and other baselines on the TinyImageNet dataset. In our experiments on the Tiny-ImageNet dataset, we also configured 50 clients with a 20% participation rate. The Dirichlet distribution parameter $\alpha$ is set to 0.3. In the Pathological partition, each client retained 80 categories, while the rest of the settings remained consistent with those for CIFAR10 and CIFAR100. Hyperparameters are determined through a grid search. The results are as in Tab. 4, which shows that the experimental results in TinyImageNet are still consistent with the interpretation in Sec. 4.2. For Pathological distribution with ViT, the test accuracy of the pFedPT is 25.95%, the accuracy of FedAvg is 9.37% and the accuracy of the FedPer is 19.23%. However, the test accuracy of the pFedPT is not the highest in Dirichlet distribution, our explanation is that it is hard to generate stable prompts in 500 rounds due to the increased task difficulty.

**The results of using pre-trained ViT.** Due to the pre-training on large-scale datasets, pre-trained ViT has demonstrated strong performance in most downstream tasks. However, the purpose of this study is to explore the design of an effective and generic prompt method specifically for the PFL task, rather than solely aiming to enhance model performance. Nonetheless, we also conducted validation using pre-trained `vit-base-patch16-224` weights, with only the output dimension of the last classification layer modified. We set the Dirichlet distribution parameter $\alpha$ is set to 0.3, and in the Pathological partition, each client retained 5 categories. The results are shown in the Tab. 5. It can be observed that the performance of the pre-trained ViT has indeed shown a significant improvement and pFedPT can also enhance the accuracy of FedAvg in non-IID settings. However, it is important to note that the model size has increased by 85 times (#parameters 1,076,826 vs 85,806,346). Therefore, the pre-trained ViT model can be used when computational efficiency is not a concern and when the sole objective is to pursue model accuracy.

Table 5: The comparison results of using pre-trained ViT weights.

| Models | Cifar10-IID | Cifar10-Dirichlet | Cifar10-Pathological |
|--------|-------------|-------------------|----------------------|
| FedAvg | **98.74** | 96.96 | 97.15 |
| pFedPT | 98.43 | **97.10** | **97.99** |

**The results with soft visual prompts.** We incorporate soft visual prompts from Visual Prompt Tuning (VPT) (Jia et al., 2022) into pFedPT to replace the original prompts. The main difference lies in the fact that the original prompts need to be added to the original images and input together into the backbone. In contrast, the soft prompts are added to the front of the sequence of features obtained after the patch embedding layer. It should be noted that soft visual prompts can only be applied to backbones with a Transformer architecture. The results in Table 6 correspond to the Non-IID scenarios, where the Dirichlet distribution parameter $\alpha$ is set to 0.3, and each client possesses 5 categories within the Pathological partition. The table reveals that, although the accuracy of soft visual prompts may not exceed that of the original pFedPT, it still surpasses FedAvg's performance. This suggests that other types of prompts are also applicable to pFedPT, serving as client-specific knowledge to assist in completing classification tasks.

Table 6: The results of pFedPT with soft visual prompts.

| Models | Cifar10-Dirichlet | Cifar10-Pathological |
|--------|-------------------|----------------------|
| FedAvg | 53.01 | 54.98 |
| pFedPT *w/* soft visual prompt | 62.04 | 57.99 |
| pFedPT | **74.92** | **75.42** |

**The results under more challenging non-IID scenarios.** We have conducted experiments on the CIFAR10 and CIFAR100 datasets with a backbone structure of CNN, where $\alpha$ is set to 0.1 and the class possession is 20%. Tab. 7 demonstrates that pFedPT continues to outperform other algorithms, even when faced with more severe data heterogeneity.

**The necessity of a four-step implementation.** To verify the effectiveness of our proposed four-step training pipeline, we conduct experiments on the CIFAR10 and CIFAR100 dataset with Dirichlet $\alpha$ set

Table 7: The results under more challenging non-IID settings.* denotes the fine-tuning results.

| #Setting | Cifar10-Dir0.1 | Cifar10-Path(20%) | Cifar100-Dir0.1 | Cifar100-Path(20%) |
|---|---|---|---|---|
| FedAvg | 53.28 | 53.44 | 26.46 | 27.60 |
| FedAvg* | 83.44 | 89.86 | 44.96 | 41.27 |
| FedProx | 53.82 | 53.56 | 26.31 | 26.49 |
| FedProx* | 86.20 | 90.58 | 45.46 | 43.47 |
| Moon | 56.81 | 55.37 | 26.61 | 27.11 |
| Moon* | 86.38 | 87.93 | 47.04 | 44.39 |
| FedPer | 86.38 | 89.97 | 45.42 | 45.73 |
| FedRep | 86.84 | 89.91 | 47.04 | 47.71 |
| FedMTL | 85.29 | 87.69 | 44.90 | 40.60 |
| FedBABU | 87.16 | 89.84 | 47.36 | 48.26 |
| Local | 85.46 | 87.66 | 43.88 | 39.46 |
| pFedPT | **87.22** | **90.84** | **47.70** | **49.61** |

to 0.3 and the class possession is 50%, where both prompts and the backbone are jointly updated in local training using a CNN backbone structure. The results are shown in Tab. 8, which indicates that the four-step implementation achieves higher accuracy, emphasizing its necessity.

Table 8: The comparison of joint training and our proposed four-step implementation.

| #Setting | Cifar10-IID | Cifar10-Dir | Cifar10-Path | Cifar100-IID | Cifar100-Dir | Cifar100-Path |
|---|---|---|---|---|---|---|
| pFedPT-joint | 65.94 | 68.84 | 75.84 | 25.62 | 27.71 | 30.52 |
| pFedPT | **66.09** | **80.83** | **81.16** | **26.41** | **32.47** | **37.89** |

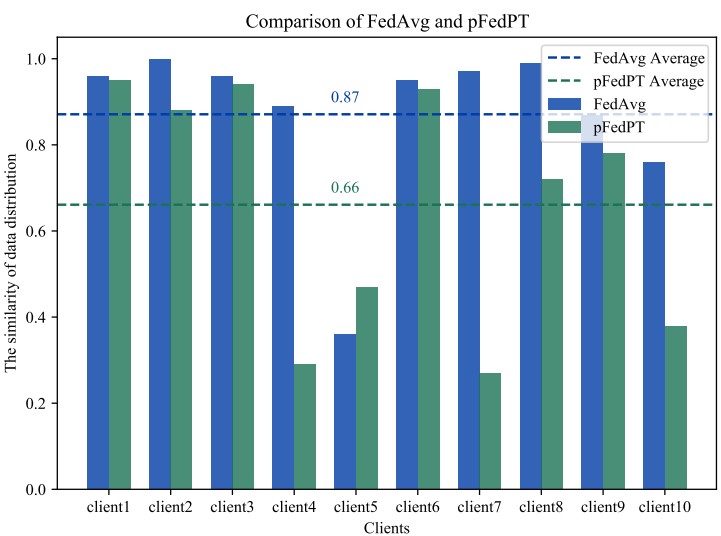

Figure 11: The results of data leakage prevention.

**The effectiveness of data leakage prevention.** Client-specific prompt information is not uploaded to the server in our method, thus preventing any additional data leakage. Furthermore, the model aggregated by pFedPT requires the loading of each client's prompt onto the image for completing the classification task. In the absence of visual prompts, the risk of inferring client-side data distribution from models uploaded to the client is significantly reduced. When testing on a balanced dataset with client-uploaded models, it becomes possible to infer the client's data distribution. Fig. 11 illustrates the similarity of 10 clients participating in the training and their inferred data distribution based on the model uploaded to the server. pFedPT reduces the predicted similarity by an average of 30% compared to FedAvg.

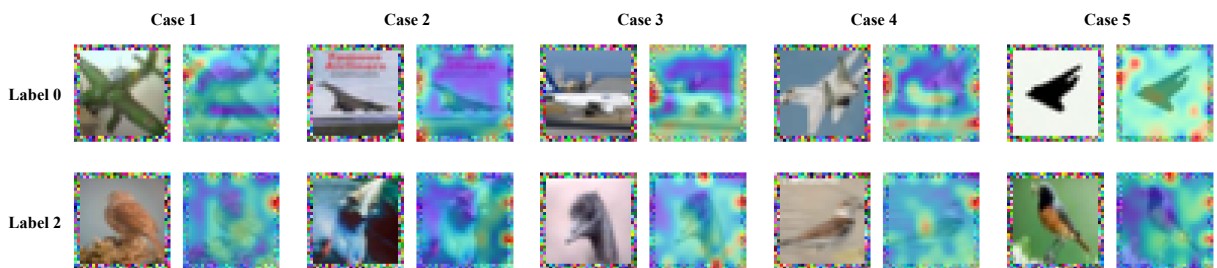

Figure 12: The attention visualization results of different samples from the same category.

**More visualization results.** We observed a phenomenon during the visualization of attention on two specific clients while processing data from the same category, as shown in Fig. 12. Specifically, we found that client 1, when processing data labeled as 0, primarily focuses on the left-middle region of the bounding box. Similarly, client 2, when processing data labeled as 2, primarily focuses on the right-middle region of the bounding box. The similarity of attention regions for the same category suggests that our prompt effectively guides local models by providing insights into the data distribution, thus facilitating accurate classification.

