# OpenReview forum: "Visual Prompt Based Personalized Federated Learning"
_TMLR — Accepted by TMLR_

### Review · Reviewer_dv1f · 2023-10-19

**Summary Of Contributions:**

This work introduced an innovative Personalized Federated Learning framework, referred to as pFedPT, for image classification tasks. pFedPT utilizes individualized visual cues to implicitly capture the local data distribution characteristics of clients, furnishing the aggregation model with additional information to enhance its classification capabilities. Comprehensive experiments demonstrate that this approach surpasses several state-of-the-art Personalized Federated Learning algorithms across diverse scenarios.

**Audience:**

Yes

**Broader Impact Concerns:**

No broader impact concerns.

**Claims And Evidence:**

Yes

**Requested Changes:**

Already listed in the weaknesses, but I have a few additional suggestions regarding the role of prompts in federated learning:
1. In Figure 5, attention maps for images of different categories are visualized, demonstrating that different images focus on different regions within the prompt. This observation is quite intriguing, and it is suggested to display the overall attention map for each category within the prompt. It might provide stronger evidence if different categories correspond to different regions.
2. Figure 6 displays t-SNE visualization of embedding for pure color images with prompts. I would like to understand how prompts affect the t-SNE visualization of the original data in each client. For example, what differences can be observed in the t-SNE results when prompts are removed from or added to the images?
3. The work should be added some discussion and experiments on representation space alignment in federated learning.

**Strengths And Weaknesses:**

### Strengths
1. The paper is well-written and very clear.
2. The concurrent works involve training prompts using a fixed pre-trained backbone. In contrast, the focus of this paper is on personalized prompt federated learning in scenarios where there is no pre-trained backbone.
3. The proposed method can be applied as an orthogonal approach to existing federated learning methods, demonstrating performance improvements across multiple datasets.
### Weaknesses
1. There is no relevant introduction to whether the use of prompts can accelerate convergence or reduce communication overhead in the paper.
2. In the described scenario, adding prompts to the input appears to primarily enhance the local data's generalization ability, and the assistance for model aggregation seems less pronounced. While the use of prompts can increase the inter-class variance of representations, there may be missing categories, as indicated in your experimental setting. For example, FedPAC[1] discussed alignment in representation spaces. However, in this paper, the addition of prompts may make it more challenging to align the representation spaces of each client.
3. RQ7 and Figure 7 appear as if adding a prompt acts as a form of noise to the client, disturbing the client's original distribution. However, this does not necessarily imply that the prompt contains information about the client's data distribution.

[1] Personalized Federated Learning with Feature Alignment and Classifier Collaboration. ICLR 2023

---

> ### Author Response · Authors · 2023-11-05
> **Response to Reviewer dv1f (1/2)**
>
> **Comment:**
>
> Thank you very much for your review and affirmation of our work. We'll answer your questions one by one in the following. We are also very honored to share some of our understandings with you.
>
> **Q1.About the question of "convergence and communication overhead"**.
>
> A1: Our method ensures the same communication load as FedAvg in every round and does not introduce any additional data transfer. Furthermore, as depicted in the graph below ([Please click here to check out the experimental results.](https://anonymous.4open.science/r/Materials_of_Response_TMLR_1532/Picture_2.png)) , pFedPT's faster convergence compared to other algorithms means that our method requires fewer communication rounds to achieve the target accuracy, thus effectively reducing the communication load.
>
>
> **Q2.About the question of "FedPAC discussed alignment in representation spaces"**
>
> A2: Your insights are indeed valuable. As indicated in the images \(c\) and (d) in Fig. 1, prompts tend to increase the inter-class variance of different sample categories on individual clients, leading to improved classification. However, this effect can indeed pose challenges in aligning the representation spaces during the aggregation process, as illustrated in Fig. 6. Even for samples from the same category, introducing different prompts can result in distinct representation spaces, which complicates the alignment process. Further consideration on how to address this issue can enhance the optimization of our method.
>
> Regarding "missing categories" on clients in a non-IID scenario, it is a normal occurrence and does not impact the above discussion and analysis.
>
> **Q3.About the question of "RQ7 and Figure 7"**.
>
> A3: The images in Fig. 7 lack the x-axis - the client data distribution. [Please click here to check out the experimental results.](https://anonymous.4open.science/r/Materials_of_Response_TMLR_1532/Picture_3.png)  In the first image, the y-axis represents the logits output obtained by inputting the client's prompt into the backbone, while in the second image, the y-axis represents the client data distribution. Fig. 7 illustrates the similarity between clients on both axes, where higher similarity is indicated by colors closer to yellow. It can be observed that the first and second images are highly similar, confirming that prompts do indeed contain knowledge related to the client's data distribution.
>
> **Q4: It is suggested to display the overall attention map for each category within the prompt**
>
> A4: We observed a phenomenon during the visualization of attention on two specific clients while processing data from the same category. [Please click here to check out the experimental results.](https://anonymous.4open.science/r/Materials_of_Response_TMLR_1532/Picture_4.png) Specifically, we found that client 1, when processing data labeled as 0, primarily focuses on the left-middle region of the bounding box. Similarly, client 2, when processing data labeled as 2, primarily focuses on the right-middle region of the bounding box. The fact that attention regions for the same category are generally similar indicates that our prompt indeed provides local models with hints about the distribution of the data, thereby assisting the models in accurate classification.
>
> **Q5: What differences can be observed in the t-SNE results when prompts are removed from or added to the images?**
> A5: We compared the impact of prompts on the t-SNE visualization of original data using the server model (no prompt generator) and a client model (with prompt generator). [Please click here to check out the experimental results.](https://anonymous.4open.science/r/Materials_of_Response_TMLR_1532/Picture_5.png) Specifically, we used 50 data samples labeled as 1 and 50 samples labeled as 4, and extracted features from the layer before the final classification for visualization. The resulting feature clustering without prompts is shown on the left-hand of the figure below while the right-hand is the features with prompts. It is noticeable that features with prompts belonging to the same category exhibit a higher degree of density, leading to reduced differences within the same class and increased differences among different classes.

---

> ### Author Response · Authors · 2023-11-05
> **Response to Reviewer dv1f (2/2)**
>
> **Q6:The work should be added some discussion and experiments on representation space alignment in federated learning.**
>
> A6: We believe that representation space alignment is reflected in the density of features obtained from encoding data of the same category, which can be supported by several experimental results:
>
> 1. The above visual comparison of features with and without prompts for two categories.
> 2. Fig.1\(c\)(d) in the original paper demonstrates that after incorporating prompts, the difference between features of the same category is significantly reduced, while the difference between different categories is increased, indicating the alignment of features in the representation space.
> 3. Fig.6 in the original paper explores whether purely prompt-based features achieve alignment in the representation space.
> 4. Fig.7 in the original paper investigates the alignment between features with prompts and the actual local data distribution.
>
> In light of the above experimental results and discussions, we believe that pFedPT can aid in aligning the representation spaces of clients with the same data category. Furthermore, aligning the representation spaces among different clients could be a promising direction for further improving our method.

---

### Review · Reviewer_y9m3 · 2023-10-20

**Summary Of Contributions:**

This paper proposes a personalized federated learning framework, pFedPT, which introduces learnable visual prompts at each client site for fine-tuning the aggregated model. The visual prompts are trained to encode the local data distribution of each client to achieve better local adaptivity. The resulting pFedPT can be integrated with existing FL and decoupled PFL methods to improve performance.
Several experiments are conducted alongside with discussing several research questions, which show that pFedPT can outperform several state-of-the-art methods.

**Audience:**

Yes

**Broader Impact Concerns:**

None.

**Claims And Evidence:**

Yes

**Requested Changes:**

Besides the weaknesses above, the following questions should be considered:
1.	Can the authors compare with other state-of-the-art methods in line with their original performance, which will make the experiment results more convincing.
2.	The research question 2 requires more evidence since the performance on CIFAR-100 is still relatively unsatisfactory.
3.	Can the work provide experimental comparison and discussion with other types of prompts, such as comparing with reference a listed in the previous question?
4.	It would be beneficial to discuss whether the learnable visual prompt will cause data leakage.
5.	It would be better to add one more dataset, such as Office-Caltech10.

**Strengths And Weaknesses:**

Strengths:
1.	The proposed framework pFedPT is novel, which first introduces the pixel patch-style visual prompt into Personalized Federated Learning frameworks.
2.	According to the experimental results and corresponding setting, the proposed method outperforms several state-of-the-art methods.
3.	The proposed method is flexible to combine with other methods.
4.	Several research questions are discussed in the experimentation, which can be used to guide further researches in the community.

Weaknesses:
1.	The writing requires improvement. Especially, citations have to be fixed throughout the manuscript.
2.	The first contribution of this paper is overclaiming. Prompting learning or prompt tuning has already been introduced in Personalized Federated Learning tasks, e.g.,
a.	Yang, F. E., Wang, C. Y., & Wang, Y. C. F. (2023). Efficient model personalization in federated learning via client-specific prompt generation. In Proceedings of the IEEE/CVF International Conference on Computer Vision (pp. 19159-19168).
b.	Guo, T., Guo, S., & Wang, J. (2023, April). pFedPrompt: Learning Personalized Prompt for Vision-Language Models in Federated Learning. In Proceedings of the ACM Web Conference 2023 (pp. 1364-1374).
In order to emphasize the contribution of this work, the authors are expected to highlight the advantage of the prompting method over others.
3.	The experimental results presented in Table 1 are different from the original paper, e.g. MOON and FedBABU, especially on CIFAR-100, they were performing much better. It would be much more convincing to refer to the original result of these works and compare with them according to their settings.

---

> ### Author Response · Authors · 2023-11-05
> **Response to Reviewer y9m3 (1/2)**
>
> **Comment:**
>
> Thank you so much for your efforts in re-evaluating our manuscript. In the following, we will respond to your concerns point by point.
>
> **Q1.About the question of "writing"**
>
> A1: Thank you for your feedback. We will further enhance the citation format and conduct an overall review of the paper's writing.
>
> **Q2.About the question of "Overclaiming of the first contribution"**
>
> A2: We will further refine our description and elaborate on the advantages of our method. Our approach is user-friendly, emphasizing the significance of prompts as prior knowledge. Furthermore, through experiments, we have validated that our method can be seamlessly integrated as a plugin with other methods, enhancing the quality of model training.
>
> **Q3.About the question of "The experimental results presented in Table 1 are different from the original paper".**
>
> A3: In Federated Learning (FL), different papers often conduct tests under varying experimental settings, which may involve different backbones, datasets, data partitions, client numbers, and participation rates. These variations can impact the final performance of algorithms, leading to differences in performance even for the same algorithm across different papers. However, as long as all methods within the same paper are compared under the same settings, the experiments are fair, and the results and conclusions are reliable.
>
> Obtaining baseline results with the original settings for all methods within the same paper can be challenging and might not be highly practical. In our paper, we set the scenario with 50 clients and a 20% participation rate, using both CNN and ViT as backbone networks. In the original MOON paper, the experimental setup involved 10 clients with full participation and ResNet as the backbone, while FedBABU used MobileNet as the network. We tested pFedPT under the aforementioned settings（still not exactly the same as the original settings）, and the results indicate that pFedPT outperforms the other two methods. The experimental results are as follows:
>
> | pFedPT  | 50.47 | 43.24 |
> | ------- | :---: | ----- |
> | Moon    | 46.30 | --    |
> | FedBABU |  --   | 40.56 |
>
> **Q4.About the question of "The research question 2 requires more evidence since the performance on CIFAR-100 is still relatively unsatisfactory" and "Add one more dataset".**
>
> A4: We have added experiments on the (Tiny-)ImageNet dataset. In our experiments on the Tiny-ImageNet dataset, we also configured 50 clients with a 20% participation rate. The Dirichlet distribution parameter $\alpha$ is set to 0.3. In the pathological partition, each client retained 80 categories, while the rest of the settings remained consistent with those for CIFAR10 and CIFAR100. Hyperparameters are determined through a grid search,  and the results are as follows:
>
>
> | (Tiny-)ImageNet Setting |  CNN-IID  | ViT-IID   | CNN-Dir   |  ViT-Dir  | CNN-Pathological | ViT-Pathological |
> | ----------------------- | :-------: | --------- | --------- | :-------: | ---------------- | ---------------- |
> | FedAvg                  |   16.22   | 15.47     | 11.01     |   12.06   | 10.53            | 9.37             |
> | FedProx                 |   15.81   | 15.53     | 10.98     |   13.82   | 11.32            | 12.84            |
> | Moon                    | **16.78** | 15.20     | 11.05     |   15.51   | 10.24            | 13.67            |
> | FedPer                  |   13.18   | 14.85     | 20.21     |   23.95   | 17.99            | 19.23            |
> | FedRep                  |   11.56   | 12.91     | 21.32     | **25.24** | 20.42            | 23.43            |
> | FedMTL                  |   11.02   | 9.75      | 17.96     |   21.14   | 17.39            | 18.30            |
> | FedBABU                 |   12.04   | 13.61     | 18.62     |   24.34   | 18.41            | 22.46            |
> | Local                   |   10.72   | 9.82      | 17.43     |   20.86   | 17.12            | 18.19            |
> | pFedPT                  |   16.53   | **18.72** | **21.42** |   21.21   | **20.66**        | **25.95**        |
>
> The experimental results demonstrate that pFedPT maintains excellent performance even on larger datasets.

---

> ### Author Response · Authors · 2023-11-05
> **Response to Reviewer y9m3 (2/2)**
>
> **Q5.About the question of "types of prompts"**.
>
> A5: Reference a uses visual prompt tokens, which are suitable for transformer structures but not applicable to network architectures like CNN. In contrast, the prompts in pFedPT can be utilized across various network structures, including CNN, making them more versatile.
>
> **Q6.About the question of "data leakage"**.
>
> A6: Client-specific prompt information is not uploaded to the server in our method, thus preventing any additional data leakage. Furthermore, the model aggregated by pFedPT requires the loading of each client's prompt onto the image for completing the classification task. In the absence of visual prompts, the risk of inferring client-side data distribution from models uploaded to the client is significantly reduced. When testing on a balanced dataset with client-uploaded models, it becomes possible to infer the client's data distribution. The table below illustrates the similarity of ten clients participating in the training and their inferred data distribution based on the model uploaded to the server. pFedPT reduces the predicted similarity by an average of 30% compared to FedAvg.
>
> | Algorithm | client1 | client2 | client3 | client4 | client5 | client6 | client7 | client8 | client9 | client10 | average |
> | --------- | :-----: | ------- | :-----: | ------- | :-----: | ------- | :-----: | ------- | :-----: | -------- | ------- |
> | FedAvg    |  0.96   | 1.0     |  0.96   | 0.89    |  0.36   | 0.95    |  0.97   | 0.99    |  0.87   | 0.76     | 0.87    |
> | pFedPT    |  0.95   | 0.88    |  0.94   | 0.29    |  0.47   | 0.93    |  0.27   | 0.72    |  0.78   | 0.38     | 0.66    |
>
>
> ##

---

### Review · Reviewer_9Pof · 2023-10-23

**Summary Of Contributions:**

This paper studies personalized federated learning (PFL). It proposes a novel approach to learning and leveraging "personalized prompts" to capture clients' characteristics to achieve promising personalized accuracy.

**Audience:**

Yes

**Broader Impact Concerns:**

No concerns.

**Claims And Evidence:**

No

**Requested Changes:**

**Major**
1. Please see the above weaknesses.

2. For the ViT backbone, I was curious why the authors did not consider adding visual prompt tokens as in Visual
Prompt Tuning (VPT) (Jia et al., 2022). In the VPT papers, adding visual prompt tokens seems to perform better than adding padding patches.

3. Can the authors compare their four-step implementation in section 3.2 with a baseline approach that jointly updates both the prompts and the backbone in local training? This will be important to justify the necessity of a four-step implementation.

**Minor**
1. Some claims are a bit too strong. Some word usage is a bit inaccurate. For example,
- *" ... which all fail to take into account the data characteristics of distributed clients."* In my humble opinion, most of the existing PFL approaches are proposed to capture the non-IID data characteristics of clients so as to perform well for each client.
- *"... prior knowledge ..."* Prompts are learned from clients' data. Can the authors provide more justification as to why they can be seen as prior?

2. The reference style could be improved. For example, *" ... both visual prompts Liu et al. (2021) and adversarial reprogramming
Elsayed et al. (2018) ..."* should be *"... both visual prompts (Liu et al., 2021) and adversarial reprogramming
(Elsayed et al., 2018) ..."* by using \citep rather than \cite.

**Strengths And Weaknesses:**

**Strength**
1. The proposed method is novel and reasonable.
2. The proposed method should be easily reproducible.
3. The paper is well-written and fairly easy to follow.

**Weaknesses**
1. The comparison to related work is not clear and informative. Specifically, the authors cited two papers using prompts in FL, PROMPTFL (Guo et al., 2022) and FedPrompt (Zhao et al., 2022). The authors listed two key differences between pFedPT and these two related works. However, the first difference --- training from scratch vs. from a pre-trained backbone --- is more like an experimental setup, not a fundamental technical difference. Could the authors provide a more direct comparison, like whether PROMPTFL and FedPrompt aim for generic FL or personalized FL?

2. The main weaknesses of the papers are in the experiments.
- The authors should not treat FedProx and MOON as PFL baselines and compare them in Table 1. These methods are not designed for PFL.
- The authors need to include one critical baseline: fine-tuning. For example, fine-tuning the global model learned in FedAvg, FedProx, and MOON. Fine-tuning has been shown to be very strong in PFL [a, b].
- The authors miss some strong PFL baselines, such as [c, d].
- The datasets are too small and limited. Please consider datasets such as (Tiny-)ImageNet.
- The backbones (both CNN and ViT) seem to differ from those used in existing works (e.g., in terms of layers). Would you be able to comment on this?
- The ViT results are a bit disappointing. In Table 1, ViT performs worse than CNN in most results. I suggest that the authors consider a pre-train ViT in the experiments.
- While the authors argued that the proposed method could be more beneficial in challenging non-IID situations, $\alpha=0.3$ and possessing $50$% classes in CIFAR are not that challenging. I suggest the authors report $\alpha=0.1$ and $20$% class possession.
- The explanation on page 10 *"... in the IID setting, we show that all the personalized solutions exhibit some extent of performance degradation, which becomes more significant as the dataset becomes more challenging. Our interpretation of this phenomenon is that when the data are distributed under the IID setting, the PFL approach does not effectively take advantage of the personalization characteristics among clients, resulting in performance degradation."* is a bit hard to grasp. First, what becomes more significant? In Table 1, all the PFL solutions improve the PFL accuracy when the datasets become more challenging (i.e., more non-IID). Second, it does not make sense to me why, under the IID setting, the PFL approach still needs to take advantage of the personalization characteristics among clients. Please note that, under the IID setting, FedAvg outperforms most of the PFL approaches.
- While Table 2 shows the compatibility, most of the results are worse than pFedPT (i.e., FedAvg + PT), which seems to suggest that pFedPT and other methods are not complementary to each other to boost the performance further. Would you be able to comment on this? Could you provide IID results in Table 2?
- The results in Fig 4 and the corresponding text are not convincing. First, for the FedAvg baseline, the authors should consider fine-tuning the whole model, not just the last layer. Second, the results are too low. The FedAvg results are merely 0.3, even lower than Local in Table 1 (both are tested on the $\alpha=30$% clients). Please note that in Table 1, even the FedAvg global model can achieve a PFL accuracy >0.5 on CIFAR-10.

**References**

[a] Chen et al., On Bridging Generic and Personalized Federated Learning for Image Classification, ICLR 2022.

[b] Yu et al., Salvaging federated learning by local adaptation. arXiv preprint arXiv:2002.04758, 2020.

[c] Shamsian et al., Personalized federated learning using hypernetworks, ICML 2021.

[d] Fallah et al., Personalized federated learning: A meta-learning approach, NeurIPS 2020.

---

> ### Author Response · Authors · 2023-11-05
> **Responses to Reviewer 8Jya (1/4)**
>
> **comment**
> Thank you very much for your review and affirmation of our work. We are very honored to share some of our understandings with you.
>
> **Q1.About the question of "the comparison to related work"**
> A1: Thank you for your feedback. We agree with your observation. The most direct comparison between pFedPT and the other two methods can be summarized as follows:
>
> * pFedPT belongs to the category of personalized Federated Learning (FL), while PROMPTFL[1] and FedPrompt fall under the generic FL category.
> * In pFedPT, the prompts are kept locally on the client-side, resulting in a unique personalized prompt for each client.
> * In contrast, PROMPTFL and FedPrompt[2] methods share prompt parameters, leading to the training of a generic FL model in the end.
>
> *[1] Guo T, Guo S, Wang J, et al. Promptfl: Let federated participants cooperatively learn prompts instead of models-federated learning in age of foundation model[J]. IEEE Transactions on Mobile Computing, 2023.
> [2] Zhao H, Du W, Li F, et al. FedPrompt: Communication-Efficient and Privacy-Preserving Prompt Tuning in Federated Learning[C]//ICASSP 2023-2023 IEEE International Conference on Acoustics, Speech and Signal Processing (ICASSP), 2023.*
>
> **Q2.About the question of "MOON,FedProx and fine-tuning baselines"**
>
> A2: Your feedback is highly valuable. Our initial consideration was that while FedProx[1] and MOON[2] may not belong to the PFL category, they serve as important baselines for addressing non-iid problems. Furthermore, our experiments later demonstrate that our method can be utilized as a plug-in to transform these two methods into PFL, thereby enhancing their performance. We will also take your suggestions seriously and provide experimental results for fine-tuning the global model learned in FedAvg, FedProx, and MOON 5 epochs here.
>
> | Cifar10 Setting      |  CNN-IID  | ViT-IID   | CNN-Dir   |  ViT-Dir  | CNN-Pathological | ViT-Pathological |
> | -------------------- | :-------: | --------- | --------- | :-------: | ---------------- | ---------------- |
> | FedAvg(fine-tuning)  | **67.12** | **60.47** | 77.52     |   72.43   | 74.95            | 67.27            |
> | Fedprox(fine-tuning) |   66.82   | 57.05     | 77.43     |   73.94   | 74.90            | 68.54            |
> | Moon(fine-tuning)    |   66.31   | 60.06     | 78.69     |   73.46   | 75.1             | 70.12            |
> | pFedPT               |   66.09   | 60.01     | **80.83** | **74.92** | **81.16**        | **75.42**        |
>
>
> | Cifar100 Setting     |  CNN-IID  | ViT-IID   | CNN-Dir   |  ViT-Dir  | CNN-Pathological | ViT-Pathological |
> | -------------------- | :-------: | --------- | --------- | :-------: | ---------------- | ---------------- |
> | FedAvg(fine-tuning)  |   25.75   | 29.97     | 32.51     |   35.88   | 31.84            | 34.79            |
> | Fedprox(fine-tuning) |   25.51   | 27.43     | 32.17     |   36.54   | 35.77            | 34.22            |
> | Moon(fine-tuning)    |   25.31   | 29.41     | 30.96     |   36.20   | 36.02            | 33.74            |
> | pFedPT               | **26.41** | **31.66** | **32.41** | **36.80** | **37.98**        | **36.88**        |
>
> We can observe that, even though fine-tuning can further improve the model's accuracy, pFedPT still outperforms these fine-tuning approaches.
>
> *[1] Li T, Sahu A K, Zaheer M, et al. Federated optimization in heterogeneous networks[J]. Proceedings of Machine learning and systems, 2020, 2: 429-450.
> [2] Qinbin Li, Bingsheng He, and Dawn Song. Model-contrastive federated learning. In Proceedings of the IEEE/CVF Conference on Computer Vision and Pattern Recognition, pp. 10713–10722, 2021a.*
>
>
>
> **Q3.About the question of "Including the fine-tuning as a critical baseline since its strong performance in PFL"**
>
> A3: As per your recommendation, we have included the experimental results for the Per-FedAvg method in reference[1], and the results are as follows:
>
> | Cifar10 Setting |  CNN-IID  | ViT-IID   | CNN-Dir   |  ViT-Dir  | CNN-Pathological | ViT-Pathological |
> | --------------- | :-------: | --------- | --------- | :-------: | ---------------- | ---------------- |
> | Per-FedAvg      |   61.35   | 57.48     | 76.81     |   68.63   | 72.32            | 74.56            |
> | pFedPT          | **66.09** | **60.01** | **80.83** | **74.92** | **81.16**        | **75.42**        |
>
> | Cifar100 Setting |  CNN-IID  | ViT-IID   | CNN-Dir   |  ViT-Dir  | CNN-Pathological | ViT-Pathological |
> | ---------------- | :-------: | --------- | --------- | :-------: | ---------------- | ---------------- |
> | Per-FedAvg       |   26.12   | 26.41     | 30.17     |   31.67   | 32.38            | 34.65            |
> | pFedPT           | **26.41** | **31.66** | **32.41** | **36.80** | **37.98**        | **36.88**        |
>
> From the table above, it is evident that pFedPT outperforms Per-FedAvg across various dataset settings.
>
> *[1] Fallah et al., Personalized federated learning: A meta-learning approach, NeurIPS 2020.*

---

> ### Author Response · Authors · 2023-11-05
> **Responses to Reviewer 8Jya (2/4)**
>
> **Q4. The currently used datasets are small and limited.**
>
> A4: As per your advice, we have included experiments on the Tiny-ImageNet datase. In our experiments on the Tiny-ImageNet dataset, we also configured 50 clients with a 20% participation rate. The Dirichlet distribution parameter $\alpha$ is set to 0.3. In the pathological partition, each client retained 80 categories, while the rest of the settings remained consistent with those for CIFAR10 and CIFAR100. Hyperparameters are determined through a grid search,  and the results are as follows:
>
> | (Tiny-)ImageNet Setting |  CNN-IID  | ViT-IID   | CNN-Dir   |  ViT-Dir  | CNN-Pathological | ViT-Pathological |
> | ----------------------- | :-------: | --------- | --------- | :-------: | ---------------- | ---------------- |
> | FedAvg                  |   16.22   | 15.47     | 11.01     |   12.06   | 10.53            | 9.37             |
> | FedProx                 |   15.81   | 15.53     | 10.98     |   13.82   | 11.32            | 12.84            |
> | Moon                    | **16.78** | 15.20     | 11.05     |   15.51   | 10.24            | 13.67            |
> | FedPer                  |   13.18   | 14.85     | 20.21     |   23.95   | 17.99            | 19.23            |
> | FedRep                  |   11.56   | 12.91     | 21.32     | **25.24** | 20.42            | 23.43            |
> | FedMTL                  |   11.02   | 9.75      | 17.96     |   21.14   | 17.39            | 18.30            |
> | FedBABU                 |   12.04   | 13.61     | 18.62     |   24.34   | 18.41            | 22.46            |
> | Local                   |   10.72   | 9.82      | 17.43     |   20.86   | 17.12            | 18.19            |
> | pFedPT                  |   16.53   | **18.72** | **21.42** |   21.21   | **20.66**        | **25.95**        |
>
> It is evident that pFedPT continues to exhibit excellent performance on the (Tiny-)ImageNet dataset.
>
> **Q5.About the question of "The backbones are different from those used in existing works"**
>
> A5: Traditional ViT models typically consist of hundreds of layers, which can lead to long training times. Therefore, we opted for a simplified version of ViT to accelerate the training process. In  FL, the specific choice of backbone network structure has a relatively minor impact on the final experimental conclusions, as long as the backbone structure is consistent across different algorithms. This ensures that the comparisons are fair and meaningful.
>
> **Q6.About the question of "Would it be worth considering pre-training ViT in the experiments to potentially improve its performance?"**
>
> A6: Due to the pre-training on large-scale datasets, Pre-trained ViT has demonstrated strong performance in most downstream tasks. However, the purpose of this study is to explore the design of an effective and generic prompt method specifically for the pFL task, rather than solely aiming to enhance model performance. Nonetheless, we also conducted validation using a pre-trained ViT model, specifically the google/vit-base-patch16-224[1] weights, with only the output dimension of the last classification layer modified. The results are shown in the table below:
>
> | Models | Cifar10-iid | Cifar10-dir0.3 | Cifar10-path0.3 |
> | ------ | :---------: | -------------- | --------------- |
> | FedAvg |  **98.74**  | 96.96          | 97.15           |
> | pFedPT |    98.43    | **97.10**      | **97.99**           |
>
> It can be observed that the performance of the pre-trained ViT has indeed shown a significant improvement  and  pFedPT can also enhance the accuracy of FedAvg  in  non-IID settins. However, it is important to note that the model size has increased by 85 times (#parameters 1,076,826 vs 85,806,346). Therefore, the pre-trained ViT model can be used when computational efficiency is not a concern and when the sole objective is to pursue model accuracy.
>
> *[1] Dosovitskiy A, Beyer L, Kolesnikov A, et al. An Image is Worth 16x16 Words: Transformers for Image Recognition at Scale. ICLR 2020.*

---

> ### Author Response · Authors · 2023-11-05
> **Responses to Reviewer 8Jya (3/4)**
>
> **Q7.About the question of " Considering the more challenging non-IID situations, like $\alpha$ = 0.1 and 20% class possession. "**
>
> A7: As per your feedback, we have conducted experiments on the CIFAR10 and CIFAR100 datasets with a backbone structure of CNN, where  $\alpha$ is set to 0.1 and the class possession is 20%. The experimental results are as follows:
>
> | Setting | Cifar10-dir0.1 | Cifar10-Pathological(20%) | Cifar100-dir0.1 | Cifar100-Pathological(20%) |
> | ------- | :------------: | ------------------------- | --------------- | :------------------------: |
> | FedAvg  |     53.28      | 53.44                     | 26.46           |           27.60            |
> | FedProx |     53.82      | 53.56                     | 26.31           |           26.49            |
> | Moon    |     56.81      | 55.37                     | 26.61           |           27.11            |
> | FedPer  |     86.38      | 89.97                     | 45.42           |           45.73            |
> | FedRep  |     86.84      | 89.91                     | 47.04           |           47.71            |
> | FedMTL  |     85.29      | 87.69                     | 44.90           |           40.60            |
> | FedBABU |     87.16      | 89.84                     | 47.36           |           48.26            |
> | Local   |     85.46      | 87.66                     | 43.88           |           39.46            |
> | pFedPT  |   **87.22**    | **90.84**                 | **47.70**       |         **49.61**          |
>
> The table above demonstrates that pFedPT continues to outperform other algorithms, even when faced with more severe data heterogeneity.
>
> **Q8.About the question of "Description of the result under the IID setting "**
>
> A8: This part of the description indeed appears unclear, and we will make revisions in the main text. For the first part of the issue, as the data's IID level increases, the performance degradation of personalized solutions compared to FedAvg becomes more significant. Regarding the second part of the issue, our focus is on assessing the performance of different algorithms in various settings, and here we mainly emphasize the IID setting. We concur with your perspective that in the IID setting, different clients inherently possess data that closely resembles the overall data distribution. As a result, many PFL methods might not be as effective, and FedAvg often outperforms most of the PFL approaches in such cases.
>
> **Q9.Further discuss for the explanation in page 10 and the extra IID results in Table 2 of original paper.**
>
> A9: Our primary focus is on the improvement that our algorithm's prompt plugin brings to the original algorithm. While other PFL methods may have their unique settings that can influence the extent of performance enhancement by the pFedPT plugin, the pFedPT plugin consistently leads to performance improvements. It can achieve accuracy comparable to the state-of-the-art (SOTA), indicating that our method can be instrumental in enhancing other algorithms. The table below presents the experimental results for the IID setting in Table 2:
>
> | Setting    | Cifar10-IID | Cifar100-IID |
> | ---------- | :---------: | ------------ |
> | FedProx    |    66.94    | 26.29        |
> | FedProx+PT |  **67.60**  | **26.87**    |
> | Moon       |    66.88    | 26.43        |
> | Moon+PT    |  **66.92**  | **26.58**    |
> | FedPer     |    51.46    | 10.82        |
> | FedPer+PT  |  **52.02**  | **11.37**    |
> | FedRep     |    49.70    | 9.13         |
> | FedRep+PT  |  **49.88**  | **9.87**     |
>
> **Q10.About the question of "The results in Fig 4 and the corresponding text are not convincing. "**
>
> A10: In general, when fine-tuning larger models like ViT, we usually adjust only the final network layers. However, as per your suggestion, we fine-tuned all the parameters of the ViT model in FedAvg, and the results are as follows. pFedPT can achieve efficiency comparable to FedAvg fine-tuning the entire model, but FedAvg requires fine-tuning a significantly larger number of parameters compared to the minimal parameter adjustment in pFedPT's prompt fine-tuning. [Please click here to check out the experimental results.](https://anonymous.4open.science/r/Materials_of_Response_TMLR_1532/Picture_1.png)
>
> Additionally, fine-tuning typically involves using a small amount of data. In our Fig. 4, the fine-tuning process, as mentioned in the paper, used only 400 samples for fine-tuning. Therefore, FedAvg's results are theoretically expected to be lower than the model accuracy trained with the entire dataset, as shown in Table 1.

---

> ### Author Response · Authors · 2023-11-05
> **Responses to Reviewer 8Jya (4/4)**
>
> **Q11.About the question of "Why the authors did not consider adding visual prompt tokens as in Visual Prompt Tuning (VPT)"**
>
> A11: The visual prompt in pFedPT serves as client-specific prior knowledge. While visual prompt tokens used in other papers are more tailored to transformer structures and might not be applicable to CNNs and other network architectures, our prompt format is more versatile and can better adapt to a variety of network structures, including CNNs.
>
> **Q12.About the question of "Justify the necessity of a four-step implementation."**
>
> A12: Following your advice, we have added experiments on the CIFAR10 and CIFAR100 dataset where both the prompts and the backbone are jointly updated in local training, with a CNN backbone structure. The results indicate that the four-step implementation achieves higher accuracy, emphasizing its necessity. The experimental results are as follows:
>
> | Setting      | Cifar10-IID | Cifar10-dir | Cifar10-Pathological | Cifar100-IID | Cifar100-dir | Cifar100-Pathological |
> | ------------ | :---------: | ----------- | :------------------: | ------------ | :----------: | --------------------- |
> | pFedPT-joint |    65.94    | 68.84       |        75.84         | 25.62        |    27.71     | 30.52                 |
> | pFedPT       |  **66.09**  | **80.83**   |      **81.16**       | **26.41**    |  **32.47**   | **37.89**             |
>
> **Q13.About the question of "capture the non-IID data characteristics"**
>
> A13: We may not have expressed it clearly, but what we meant is that  current  PFL method does not involve direct client-side data feature modification to achieve PFL.
>
> **Q14.About the question of "Can the authors provide more justification as to why they can be seen as prior?"**
>
> A14: The Experiments section, particularly RQ5, RQ6, and RQ7, provides more detailed answers to your questions. Experiments related to RQ6 show that prompts for different clients are distinct and related to specific client characteristics. The results from RQ7 further affirm that prompts contain information about the client's data distribution. The experiments in RQ5 demonstrate that the information encoded in the prompts can indeed be attended to by the backbone. These experiments collectively indicate that prompts can be regarded as priors related to the client's data distribution, are attended to by the backbone, and thereby provide relevant knowledge about the client's data distribution to assist in the training task.
>
> **Q15.About the question of "reference style"**
>
> A15: Thank you for your feedback, and we will consider your suggestions for revisions in the next version.

---

> ### Comment · Reviewer_9Pof · 2023-12-04
> **Reviewer's feedback to the rebuttal**
>
> I appreciate the authors' extreme effort in preparing the responses. They address most of my concerns. I have some more questions/suggestions about Q5, Q6, Q7, Q8, Q10 (main feedback), and Q14. I would hope that the authors can address them in the final version to strengthen the paper and remove inaccurate claims.
>
> **Q1:** Thank you. Please incorporate this discussion into your final version.
>
> **Q2.** Thank you. Please incorporate the fine-tuning results into your final version. In my humble opinion, fine-tuning is the essential baseline of personalized FL. Even though the gain of your approach reduces when compared to fine-tuning, you may claim some additional advantages like users do not need to store entire personalized models but their prompts.
>
> **Q3.** Again, please include the results in your final version.
>
> **Q4.** Thank you. Please incorporate the results into your final version. Please also include those suggested references.
>
> **Q5.** Thank you for the explanation. Can you authors confirm whether they use the same model architecture for all the other compared methods?
>
> **Q6.** Thanks. I would strongly suggest that the authors apply the pre-trained models to more challenging datasets like Tiny-ImageNet.
>
> **Q7.** Thanks. In the final version, for all these comparisons to generic FL methods (like MOON), the authors should include fine-tuning results. Otherwise, the results are misleading and not fair.
>
> **Q8.** Please modify your claims accordingly in the final version. In my humble opinion, your focus is on personalized FL. Thus, there is probably not much need to discuss the IID cases. It is better to remove some inaccurate claims.
>
> **Q9.** Thanks. Please include the results in your final version.
>
> **Q10.** Thank you for the response. I understand your argument. However, the authors should really use full fine-tuning for FedAvg as it has a huge gain in performance (as the authors' new results show). Fine-tuning only the last layer is obviously sub-optimal and misleading.
>
> *Additionally, fine-tuning typically involves using a small amount of data. In our Fig. 4, the fine-tuning process, as mentioned in the paper, used only 400 samples for fine-tuning. Therefore, FedAvg's results are theoretically expected to be lower than the model accuracy trained with the entire dataset, as shown in Table 1.*
>
> I am not sure if the authors got my original comments about the poor performance. The new clients the authors considered are sampled from CIFAR. If the new clients do not have enough data, the naive solution is to directly apply the FedAvg global model (trained from other participating clients) to new clients. According to Table 1, the result should probably be around 50%. In Fig 4, the FedAvg's result is just 30%. The authors may either consider just using the global model without fine-tuning, or full fine-tuning, rather than fine-tuning the last layer which leads to a very low performance. Please also note that, in Table 1, each client roughly has 1K training examples (more than 400 but still quite limited), but the local performance can already achieve 68.4. I think to make Fig. 4 more meaningful, direct local training with the 400 new client samples is needed. I would expect the result to be higher than 30%.
>
> Overall, my comment about Fig. 4 is just that the FedAvg result does not make sense from multiple perspectives. The gap between pFedPT and FedAvg seems to be a bit exaggerated. To be clear, even without Fig. 4 and the corresponding experiments, the paper already has sufficient results and contributions. I just hope the results and claims to be included in the final version are accurate and convincing.
>
> **Q12.** Thank you. Please incorporate the results into the final version.
>
> **Q14.** Thank you. However, in my humble opinion, the prompts are learned from clients' data and probably should not be named *priors*. I would suggest the authors change to another term.

---

> > ### Author Response · Authors · 2023-12-06
> > **Responses to Reviewer 8Jya**
> >
> > ## **Responses to Reviewer 8Jya**
> >
> > Thank you very much for your response. We will incorporate the previously discussed content and relevant results into the final version, which will be uploaded shortly. Regarding the questions/suggestions for Q5, Q6, Q7, Q8, Q10, and Q14, we would like to offer the following insights:
> >
> > **Q5.** In our experiments, we employed two types of model architectures as backbones: CNN and a simplified version of ViT. When comparing different methods, we ensured that the only variations were in the aggregating and updating methods, while keeping the dataset and the utilized backbones consistent, thereby ensuring a fair comparison.
> >
> > **Q6.** Thank you for your suggestion. We will apply pre-trained models to Tiny-ImageNet to observe the experimental results. However, this may take some time. We will present the relevant results in subsequent responses and in the final version when the experiments are finished.
> >
> > **Q7.** Thank you for your suggestion. The fine-tuning results will be incorporated in the final version when the experiments are finished.
> >
> > **Q8.** We greatly appreciate your suggestion. We will rephrase this section, removing some inaccurate claims, to avoid causing confusion among readers.
> >
> > **Q10.** Firstly, we agree with your point that using full fine-tuning for FedAvg can result in a significant performance gain.
> >
> > We would like to clarify that in Fig.4, FedAvg is trained on clients with a data distribution of Dir(0.1) and fine-tuned on clients with a distribution of Dir(0.3). The fine-tuning process does not involve model aggregation and, due to the limited number of samples, the training accuracy is theoretically lower than the 85.46 accuracy of local training in the Q7 discussion.
> >
> > As for the reasons behind the poor performance, we attribute it to the disparity in data distribution between model training and fine-tuning. Tab.1 does not address similar configurations, making it challenging to draw meaningful conclusions about this situation from the results in Tab. 1. We also concur with your suggestion of using full fine-tuning or local training as a baseline for Fig.4.
> >
> > Considering the insights you provided, we acknowledge that the experimental details and corresponding descriptions in Fig. 4 may lead to confusion and may not add significant value. To enhance readability for readers, we plan to remove this discussion in the final version.
> >
> > **Q14.** At first, we believed that the prompt learned from client data contained certain client-specific information, considering it akin to prior knowledge for the backbone. However, based on your suggestion, we would like to refer to this aspect as 'client bookmarks.' This term signifies the learning of client-specific knowledge that can assist the backbone in distinguishing between different clients.

---

> > ### Author Response · Authors · 2023-12-16
> > **Additional Responses to Reviewer 8Jya**
> >
> > We supplement the experimental results mentioned in Q6 and Q7 and are actively preparing the final version. We anticipate completing the final version by the end of this week.
> >
> > **Q6.**
> > We conduct additional experiments applying pre-trained models to Tiny-ImageNet. The results align with those obtained in Cifar10 experiments, indicating a significant improvement in the performance of the pre-trained ViT. Furthermore, pFedPT demonstrates the ability to enhance FedAvg accuracy in non-IID settings.
> >
> > | Models   | Tiny-ImageNet-iid | Tiny-ImageNet-dir0.3 | Tiny-ImageNet-path0.3 |
> > |----------|:------------------:|----------------------|-----------------------|
> > | FedAvg   | **80.42**           | 74.37                | 69.01                 |
> > | pFedPT   | 79.77              | **77.23**             | **75.53**              |
> >
> > **Q7.**
> > We provide additional details on the results for FedAvg, FedProx, and Moon under the fine-tuning setting. Fine-tuning significantly improves accuracy. For instance, after applying fine-tuning on the Cifar10-dir0.1 dataset, the accuracy of FedAvg increases from 53.28% to 83.44%. However, it still falls short of the accuracy achieved by pFedPT, which is 87.22%. This finding corresponds with our conclusion in aforementioned response to Q2.
> > | Setting                   | Cifar10-dir0.1 | Cifar10-Pathological(20%) | Cifar100-dir0.1 | Cifar100-Pathological(20%) |
> > |---------------------------|:--------------:|---------------------------|:---------------:|-----------------------------|
> > | FedAvg |  53.28 |53.44|26.46  |27.60|
> > | FedAvg (fine-tuning)      | 83.44          | 89.86                     | 44.96           | 41.27                        |
> > | FedProx |  53.82 |53.56|26.31  |26.49|
> > | FedProx (fine-tuning)     | 86.20          | 90.58                     | 45.46           | 43.47                        |
> > | Moon |  56.81 |55.37|26.61  |27.11|
> > | Moon (fine-tuning)        | 86.38          | 87.93                     | 47.04           | 44.39                        |
> > | pFedPT                    | **87.22**      | **90.84**                 | **47.70**       | **49.61**                    |

---

### Decision · Action_Editor_zHNM · 2023-12-14

**Recommendation:** Accept with minor revision

**Comment:**

The paper focuses on personalized federated learning, and addresses an issue of not considering the data distributions of distributed clients, which is overlooked by existing methods. In the scenario of image classification, the paper proposes a framework that leverages personalized visual prompts to implicitly represent local data distributions of clients; this prior knowledge of local distributions from visual prompts can also be used during testing. Experiments on CIFAR datasets show the efficacy of the proposed framework.

While reviewers generally acknowledge the novelty of introducing visual prompts into the PFL problem, they have concerns on the writings (e.g., overclaiming of contributions, reference formats, etc), on the settings of experiments to compare with existing methods, on the less thorough designs of the experiments, and on discussions of the downside effects by using visual prompts into PFL. To make fair claims of the contributions, the authors are expected to emphasize the specific, technical advantages proposed in this submission; the authors are also highly encouraged to compare different methods under one or several same settings - although in the responses the authors explained that this could be challenging, however, if a fair and consistent protocol of comparisons cannot be established for PFL topic, the topic itself could vanish eventually. The authors would further revise the paper by inclusion of fine tuning as a critical baseline, and by inclusion of other stronger PFL baselines. Some of these revisions have already been done, while others need to be conducted and included in the revised paper. In addition, in the revised version, the authors are expected to provide experiments and discussions with other types of prompts.

**Audience:**

Yes

**Claims And Evidence:**

Yes (after the review and response phase)

---

> ### Author Response · Authors · 2024-01-16
> **Responses to  Editor zHNM**
>
> **Response to Action Editor zHNM**
>
> Dear AE zHNM,
>
> Thank you for your efforts to enhance the quality of our manuscript. We appreciate the positive feedback on the novelty of the paper. We have carefully considered the issues raised by the reviewers and have made necessary revisions to address them adequately.
>
> **About writings**
> We have made corrections to the writing issues raised by the reviewers, rephrased our contributions at the end of the introduction, and corrected the citation format throughout the entire manuscript.
>
> **About the settings of experiments**
> In response to the reviewer's inquiry regarding the experimental settings section, we have incorporated the outcomes of the Per-FedAvg method suggested by reviewer 9Pof into Tab. 1. Additionally, we have enriched Appendix A.2 with results stemming from fine-tuning based on FedAVG, FedProx, and Moon, thereby substantiating the efficacy of pFedPT on the TinyImageNet dataset. Addressing the concern about "comparing different methods under one or several same settings," as highlighted by the reviewer, it is noteworthy that not all PFL studies employ the same datasets and network structures. Nevertheless, the majority of these studies, including our own, adhere to a consistent set of parameters when comparing various methods within the same paper. This meticulous comparison ensures both the fairness of experiments and the reliability of results and conclusions.
>
>
> **About the discussions with other types of prompts and downside effects by using visual prompts**
> We have analyzed various options for the placement of prompts in input images and explored their impact on the model performance, presenting the relevant results in Fig. 8. Additionally, in the experiments discussed in Tab. 6 of the appendix, we have examined the effectiveness of soft visual prompts for pFedPT in Non-IID scenarios, indicating that other types of prompts are also applicable to pFedPT, providing client-specific knowledge to assist in completing classification tasks. The results of Table 6 are as follows:
>
> | Models | Cifar10-Dirichlet | Cifar10-Pathological |
> | -------- | -------- | -------- |
> | FedAvg | 53.01 | 54.98     |
> | pFedPT with soft visual prompt | 62.04 | 57.99     |
> |  pFedPT  | **74.92** | **75.42**  |
>
> Regarding the downside effects,  our experiments in Fig. 5 highlight that the inclusion of visual prompts leads to a distinctive representation space for each client, requiring the backbone to exert more effort for alignment on the server.
>
> Finally, it is worth noting that we have released the source code for our proposed methods to the research community. This commitment ensures the reproducibility of our work.
>
> We acknowledge the AE's overall assessment of our paper. We are confident that these revisions and new features will strengthen the manuscript and address the concerns raised by the reviewers. We appreciate the opportunity to improve our work and look forward to the final decision on our submission.
>
> Thank you for your time and consideration.
>
> Sincerely,
>
> Authors